Corrected: Publisher correction

# Contemporaneous 3D characterization of acute and chronic myocardial I/R injury and response

Simon F. Merz[1,2,8], Sebastian Korste[3,8], Lea Bornemann 🄳 [1,8], Lars Michel[3], Pia Stock[3], Anthony Squire[1], Camille Soun[1], Daniel R. Engel[1], Julia Detzer[4], Holger Lörchner[4,5], Dirk M. Hermann[6], Markus Kamler[7], Joachim Klode[2], Ulrike B. Hendgen-Cotta[3], Tienush Rassaf[3], Matthias Gunzer 🄳 [1,9] & Matthias Totzeck 🄳 [3,9]

Cardioprotection by salvage of the infarct-affected myocardium is an unmet yet highly desired therapeutic goal. To develop new dedicated therapies, experimental myocardial ischemia/reperfusion (I/R) injury would require methods to simultaneously characterize extent and localization of the damage and the ensuing inflammatory responses in whole hearts over time. Here we present a three-dimensional (3D), simultaneous quantitative investigation of key I/R injury-components by combining bleaching-augmented solvent-based non-toxic clearing (BALANCE) using ethyl cinnamate (ECi) with light sheet fluorescence microscopy. This allows structural analyses of fluorescence-labeled I/R hearts with exceptional detail. We discover and 3D-quantify distinguishable acute and late vascular I/R damage zones. These contain highly localized and spatially structured neutrophil infiltrates that are modulated upon cardiac healing. Our model demonstrates that these characteristic I/R injury patterns can detect the extent of damage even days after the ischemic index event hence allowing the investigation of long-term recovery and remodeling processes.

[1] Institute for Experimental Immunology and Imaging, University Hospital, University Duisburg-Essen, 45147 Essen, Germany. [2] Department of Dermatology, Venerology and Allergology, University Hospital Essen, 45147 Essen, Germany. [3] Department of Cardiology and Vascular Medicine, University Hospital Essen, 45147 Essen, Germany. [4] Dept. of Cardiac Development and Remodelling, Max Planck Institute for Heart and Lung Research, 61231 Bad Nauheim, Germany. [5] German Centre for Cardiovascular Research (DZHK), Partner site Rhine-Main, Frankfurt am Main, Germany. [6] Department of Neurology, University Hospital Essen, 45147 Essen, Germany. [7] Department of Thoracic and Cardiovascular Surgery, University Hospital Essen, 45147 Essen, Germany. [8] These authors contributed equally: Simon F. Merz, Sebastian Korste, Lea Bornemann. [9] These authors jointly supervised this work: Matthias Gunzer, Matthias Totzeck. Correspondence and requests for materials should be addressed to M.G. (email: Matthias.Gunzer@uni-due.de) or to M.T. (email: Matthias.Totzeck@uk-essen.de)

Restoration of perfusion after acute myocardial infarction is the most frequent and effective medical treatment, but the process may inflict massive ischemia/reperfusion (I/R) injury[1]. A hallmark of I/R injury is the major loss of the vasculature[2] accompanied by impairment of the endothelium. In parallel, I/R results in an extensive inflammatory response[3]. Within minutes, circulating neutrophils are attracted to the affected vasculature where they transmigrate into the damaged tissue with incompletely defined roles in injury progression or protection[4,5]. Macrophages, in turn, exhibit complex functions in cardiac inflammation and injury site repair peaking days after the onset of I/R[6]. A precise spatial identification and characterization of the cellular composition of the injured region reflecting the complex interplay between I/R-signaling and repair mechanisms, therefore, is essential for their breakdown into tractable therapeutic targets[7].

The established tools for the characterization of I/R injury and response assessment have several limitations. This particularly relates to the co-localization of pathophysiological events including inflammation and (re-) vascularization. The mainstay for cardiac injury analysis in animal models is measurement of metabolic activity of cardiomyocytes in serial thick sections using triphenyl tetrazolium chloride (TTC)[8]. However, this does typically not include advanced histological and immuno-histochemical co-assessments and fails to provide an accurate 3D reconstruction of the affected tissue. To precisely localize and quantify the extent of cardiac damage, a global 3D-structural analysis of the heart is mandatory. Light sheet fluorescence microscopy (LSFM) has been particularly effective to perform similar analyses and was previously used to image large tissue specimens[9,10], including the heart[11]. Therefore, we hypothesized that LSFM was capable of characterizing myocardial ischemia/reperfusion (I/R) injury in conjunction with significant I/R injury response mechanisms, particularly immune cell infiltration. However, the technique requires high tissue transparency. In murine organs, this can be achieved by complex, often toxic chemical pretreatments, collectively termed clearing[12,13].

We have recently introduced a non-toxic clearing method using ethyl cinnamate (ECi) as the refractive index-matching component[9]. While ECi performed well for murine organs, such as kidney and bone, it suffered, like other approaches, from extremely high tissue autofluorescence of the heart muscle. Until now, this made quantitative LSFM of whole hearts challenging.

Hence, we wished to (i) develop a readily applicable, non-toxic workflow with chemicals and tools that are commercially available, (ii) benchmark this work flow against standard I/R histology techniques in compliance with recent guideline recommendations[14], (iii) quantify I/R injury parameters in 3D, (iv) assess the long-term impact of I/R injury following days after reperfusion and (v) relate the I/R injury zones to immune response mechanisms.

Here we introduce an ECi-based 3D myocardial I/R injury assessment workflow, termed BALANCE (Bleaching-Augmented soLvent-bAsed Non-toxic ClEaring), to overcome the limitations of the current analysis tools.

## Results

### BALANCE enables 3D whole heart imaging and analysis.
Central to BALANCE is a peroxide-based bleaching step compatible to antibody-mediated fluorescence labeling in vivo. Bleaching was key to unlock lower-wavelength channels for homogenous imaging (Fig. 1 and Supplementary Figure 1a and d). We tested other established protocols[15] for tissue autofluorescence reduction and found BALANCE to be the most suitable for homogenizing tissue autofluorescence fast and throughout the whole heart using our I/R injury protocol workflow (Supplementary Figure 1, Supplementary Table 1 and Supplementary Note 1). Applying BALANCE to our samples, tissue autofluorescence itself could be used to visualize the overall heart muscle structure allowing clear detection of cardiomyocytes and their nuclei (Fig. 2a, b) at much higher optical resolution than previously possible[16,17].

CD31, a surface molecule on endothelial cells, is a potential marker for vascular integrity[18]. Intravital staining with fluorescent anti-CD31 antibodies[9] revealed homogenously labeled vasculature- and heart-structures as shown before[19,20] (Fig. 2a). BALANCE now allowed to combine autofluorescence and CD31 signals to precisely reconstruct prominent but delicate structures such as the aortic valve (Fig. 2c and Supplementary Movie 1). Additionally, we could demonstrate the applicability of our clearing protocol to other murine tissues (e.g. liver, Fig. 2d)[9,21,22] and the human heart (left atrial appendage biopsy, Fig. 2e). Note that BALANCE was designed to maximize the signal quality of intravenous (i.v.)-mediated staining using synthetic fluorophores, allowing cell identification with high precision against a highly autofluorescent background. However, homogenizing tissue autofluorescence with peroxide may quench endogenously expressed fluorophores (Supplementary Figure 2, Supplementary Note 2).

### Generating infarct and area at risk volumes for I/R analysis.
Next, we identified the 3D-extent of I/R injury in a model of left coronary artery (LCA) I/R. This induced an area of terminated blood supply, which could be restored upon reperfusion (area at risk [AAR])[23]. To precisely define and quantify the AAR, we injected a fluorescein isothiocyanate (FITC)-conjugated albumin solution into the aorta following re-occlusion of the LCA in the excised heart. This led to a clearly detectable AAR as FITC negative (FITC$^{neg}$), hence non-perfused zone (Fig. 3a, b). I/R generated an AAR that comprised around 29% (21–35, median and interquartile range, $n = 6$) of the whole heart (Fig. 3b, Supplementary Movie 2, Supplementary Figure 5e).

The onset of I/R injury causes not only a deterioration of cardiomyocytes but also neighboring endothelial cells[18]. Therefore, this I/R injury could be identified by complete loss of CD31 signals, termed CD31$^{neg}$, in LSFM images (Fig. 3c). 3D tracing of CD31$^{neg}$ areas allowed to generate volumetric and quantifiable 3D infarct-bodies (Fig. 3c, Supplementary Movie 3). These were clearly distinguishable from autofluorescent artifacts showing reduced CD31 signal (CD31$^{dim}$) which were seen in all hearts (Supplementary Figure 3). Figure 3b, c shows a CD31$^{neg}$ to AAR injury of 10%, but the spectrum of small to larger infarction of this model can be visualized as outlined below. Furthermore, 3D vessel tracing allowed to reconstruct the relation between vessels affected by I/R, the ensuing AAR and the respective size and localization of infarct-bodies (Fig. 3d). This analysis also demonstrated that vessels occluded during I/R matched the AAR defined by FITC-absence (Supplementary Movie 4).

### Benchmarking of CD31-based I/R injury analysis.
Histological 2D approaches using triphenyl tetrazolium chloride (TTC) have been the mainstay for quantifying the degree of I/R injury[24]. Cardiomyocytes with intact mitochondria convert TTC to a red dye, while I/R-affected cells remain unstained (TTC$^{neg}$). By classical TTC staining of sequential 2-mm slices, we assessed the I/R injury. When we analyzed the same slices by LSFM, we indeed observed CD31$^{neg}$ regions that overlapped with TTC$^{neg}$ areas (Fig. 4 and Supplementary Figure 4). While larger vessels appeared to retain CD31 inside the infarcted area, smaller capillaries had completely lost their CD31 signal. The 3D

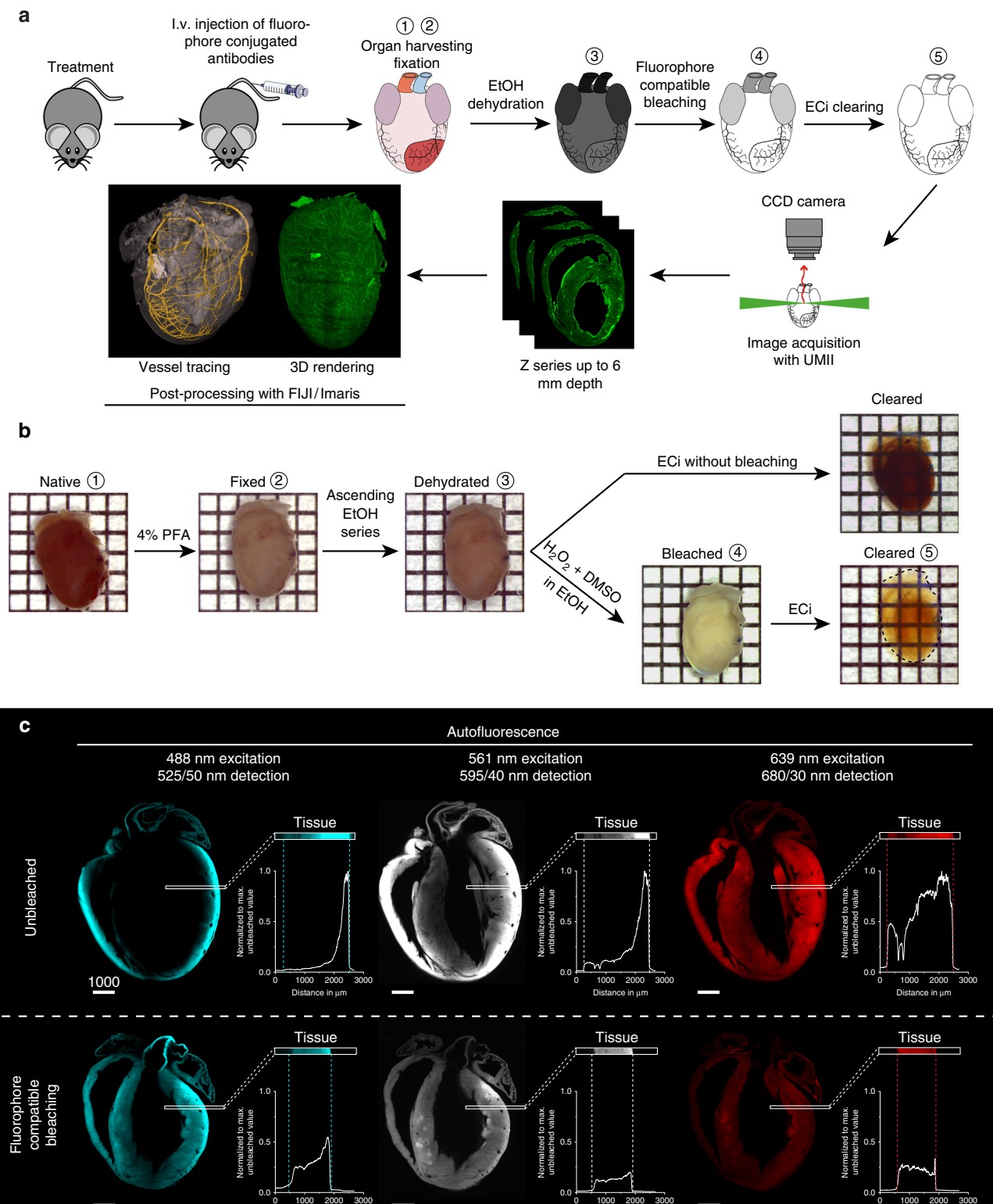

**Fig. 1** Fluorophore compatible bleaching enables homogenous light sheet fluorescence microscopy (LSFM) in whole murine hearts. **a** Graphical abstract of light sheet guided heart analysis procedure. **b** Macroscopic images of murine hearts during protocol steps. In comparison to the non-bleached sample, fluorophore compatible bleaching enhances the clarity of the sample. One square equals 2 × 2 mm. **c** Signal distribution of ethyl cinnamate (ECi)-cleared, unbleached or bleached hearts in LSFM in different channels, single wavelength excitation and filtered detection range given in wavelength/range, detecting autofluorescence. The region of interest (ROI) in the left ventricular wall (LVW) shows high autofluorescence at the edge of unbleached hearts. This peak is lowered, together with an overall lower autofluorescence intensity in bleached hearts. The homogenization of autofluorescence is most prominent in the short wavelength channels, unlocking those for quantitative imaging and general thresholding. Scale bar values in µm. Source data are provided as a Source Data file

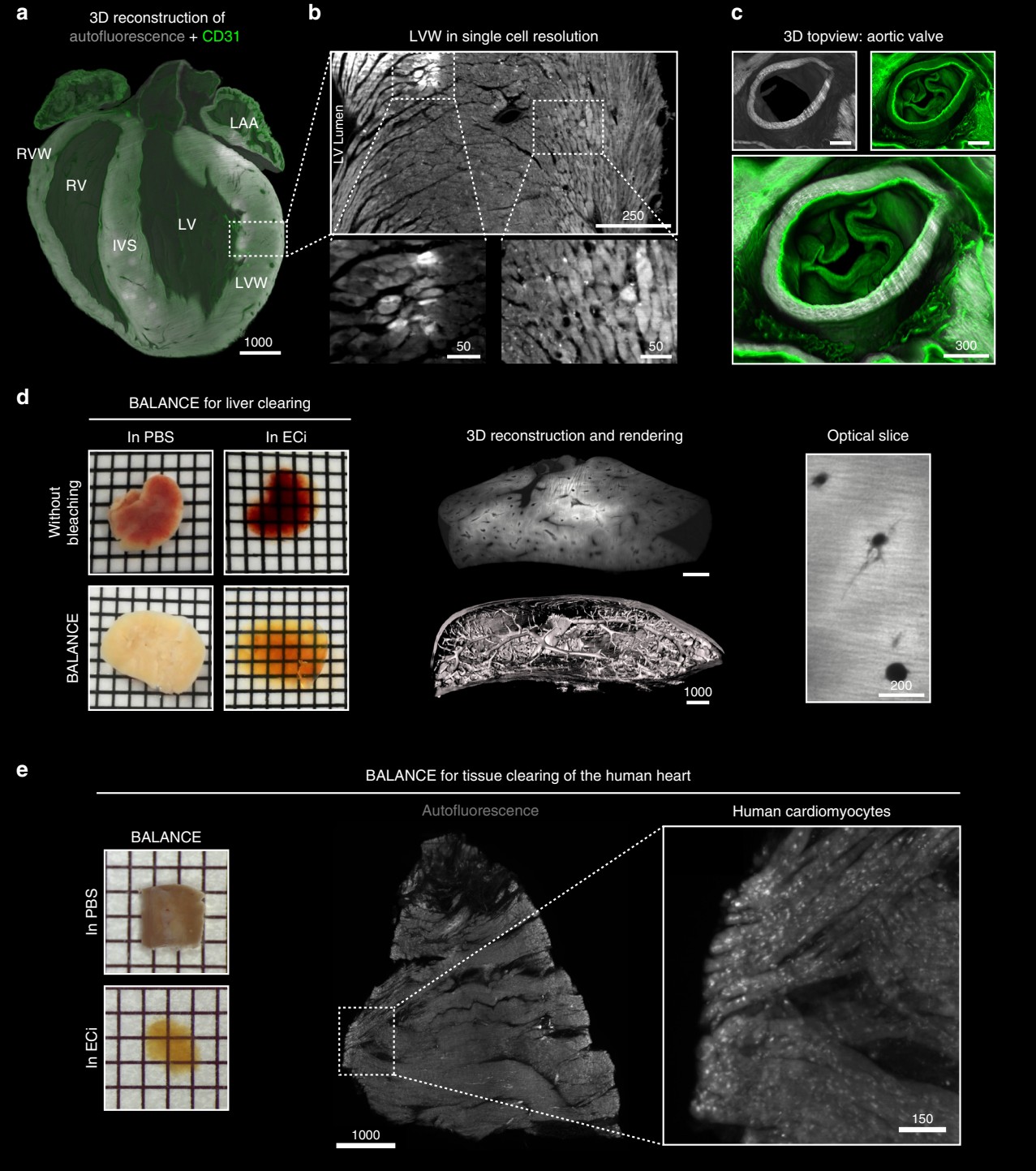

**Fig. 2** High resolution light sheet fluorescence microscopy (LSFM) of murine and human tissue. **a** 3D reconstruction of the inferior (dorsal) half of an ECi-cleared murine heart depicting both CD31 (blood vessels; green) and autofluorescence (total heart tissue; grey) showing distinct heart structures such as right ventricular wall (RVW), right ventricle (RV), intraventricular septum (IVS), left ventricle (LV), left ventricular wall (LVW) and left atrial appendage (LAA). **b** Single optical slice obtained by LSFM showing a cross-section of the LVW (enlarged ROI highlighted in **a**) with cellular detail using autofluorescence (grey) only. Enlargements show that high autofluorescence in restricted regions is of cellular origin (ROI left, different contrast settings) and that the directionality of visualized cardiac cells depends on their respective localization within the muscle (ROI right). Magnification: 8 ×. **c** 3D reconstruction of the aortic valve in a top-view visualized by autofluorescence (grey) and CD31 (green). Magnification: 4 ×. **d** BALANCE applied to murine liver enhances sample transparency (left) and enables imaging of large tissue samples for quantitative purposes (mid and right). High contrast between vessel lumen and surrounding tissue allows direct rendering of the portal vein system from autofluorescence raw data (mid). **e** Left: BALANCE also enables clearing of a human left atrial appendage (LAA) biopsy and enhances sample transparency. Mid: The autofluorescence signal alone gives a structural overview and single cellular resolution (right). Scale bar values in μm. One square on the macroscopic images equals 2 × 2 mm. (ECi - ethyl cinnamate)

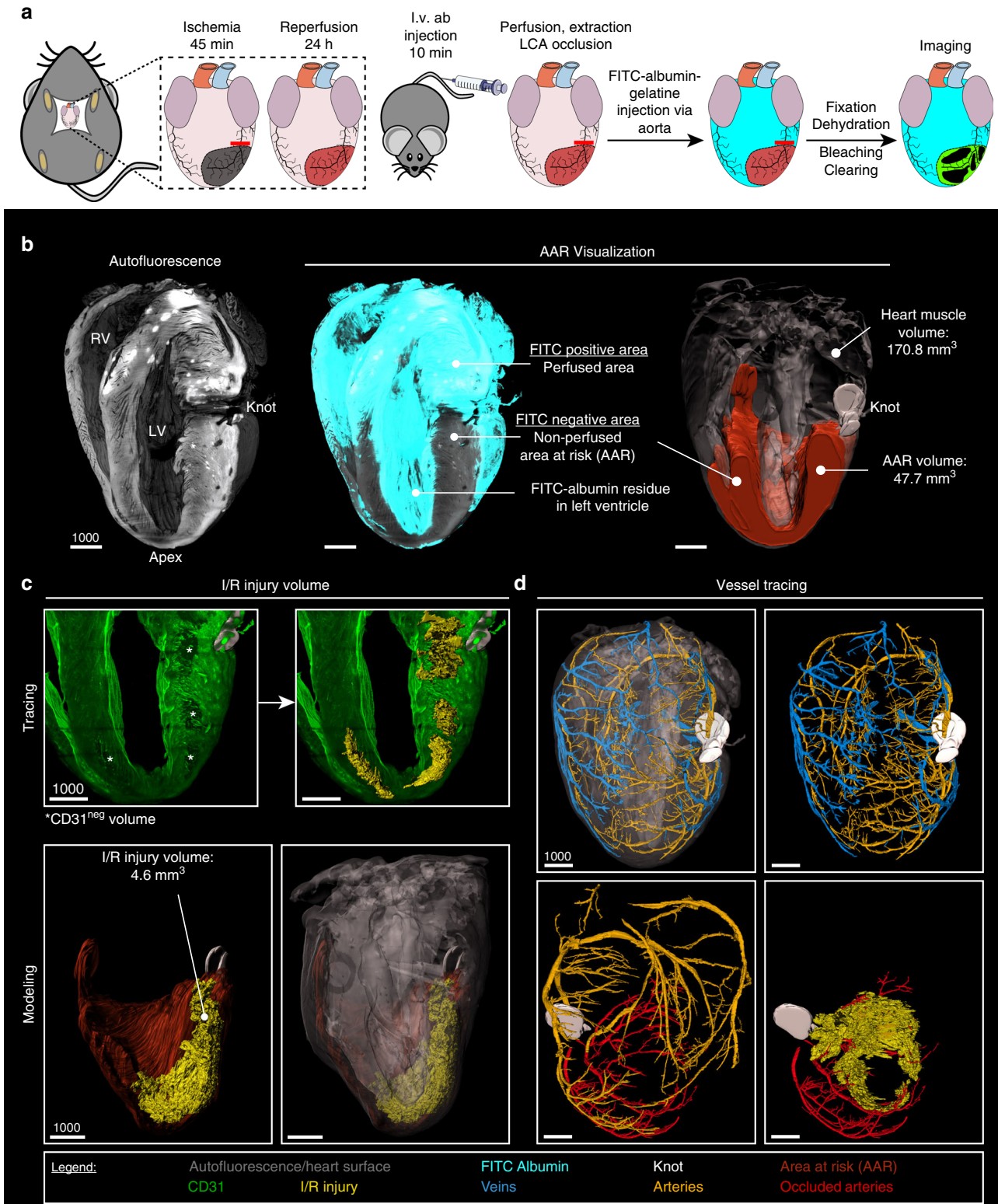

**Fig. 3** 3D quantification and characterization of ischemia/reperfusion (I/R) injury size using light sheet fluorescence microscopy (LSFM). **a** Workflow for LSFM-guided 3D analysis of I/R injury. **b** Quantification in 3D; Left: heart model based on autofluorescence (grey) for orientation. Middle: an overlay with the fluorescein isothyocyanate (FITC)-albumin filling (turquoise), all perfused areas are FITC positive (FITC$^{pos}$), whereas non-perfused areas show no FITC signal (FITC$^{neg}$) and autofluorescence only. Right: traced FITC$^{neg}$ area, building up a volume model for the area at risk (AAR, orange). The traced shape of the knot (white) together with the heart surface (translucent grey) give detailed spatial information and enable volume measurements. **c** I/R injury quantification; Top: based on slice by slice tracing of areas negative for CD31 (CD31$^{neg}$) (white stars, left) a volume model of I/R injury (right) is reconstructed (bright yellow). Bottom: combined 3D models of I/R injury (yellow), AAR (orange), knot (white; left) and heart surface (right). **d** In vivo CD31 labeling and spatial information allow tracing of major arteries (dark yellow) and veins (blue). Together with the visualization of the knot, occlusion of` the left coronary artery (LCA) can be confirmed and occluded arteries can be identified (red). Scale bar values in μm

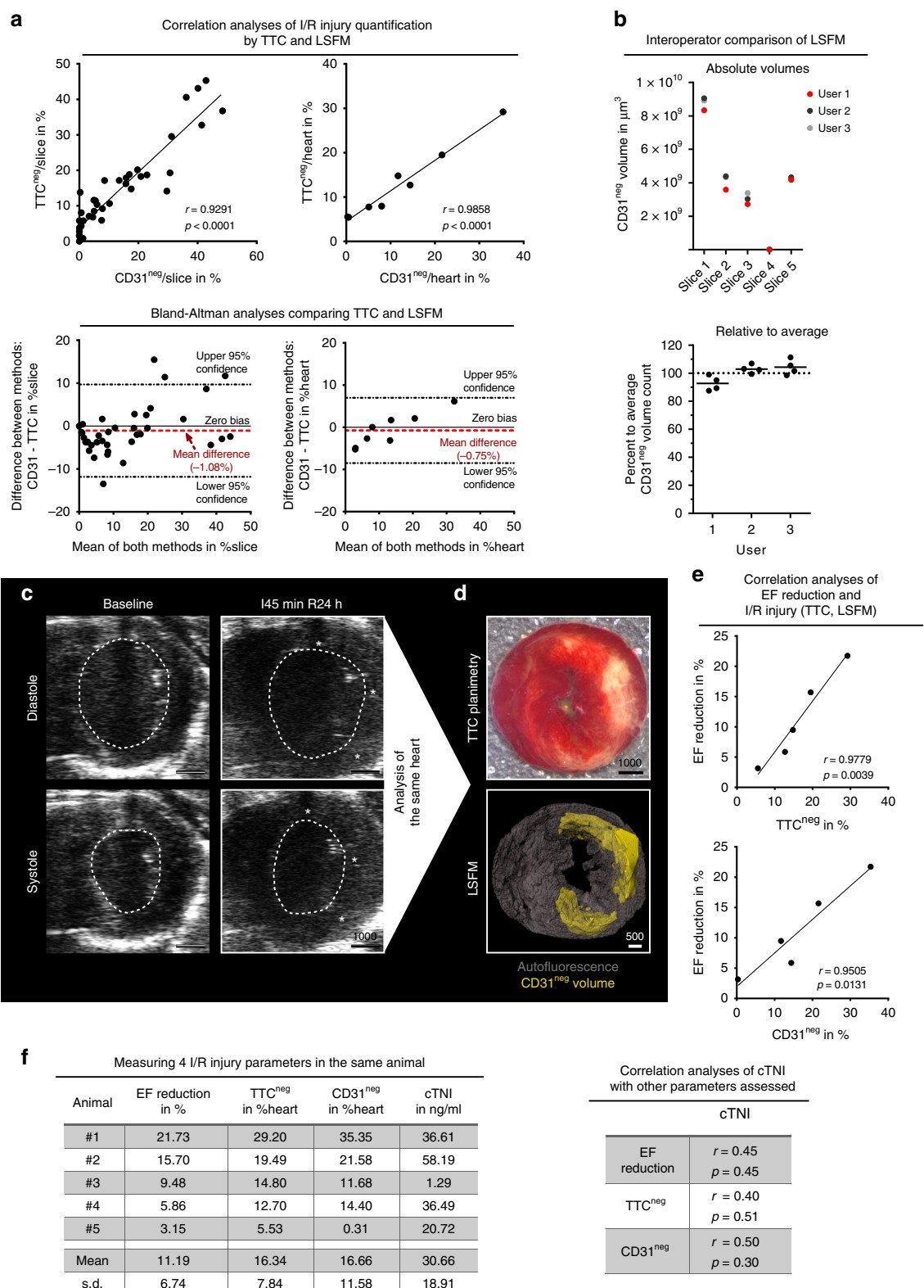

analysis also demonstrated that TTC overshadowed the fine structural details of I/R damage. In contrast, LSFM of CD31 faithfully reconstructed the complex 3D structure of the I/R injury-site after 24 h without destroying the tissue (Supplementary Figure 4). We correlated CD31[neg] volumes to standard markers of myocardial damage, such as TTC staining, ejection fraction (EF) reduction and cardiac troponin I (cTNI) release, as recommended tools in experimental studies[14]. We found a highly

linear relationship between TTC[neg] areas, CD31[neg] volumes (both normalized to single slices as well as total hearts) and EF reduction (Fig. 4a, e). Regions of abnormal left ventricular wall movement were located mostly in mid anterior, mid lateral and mid inferolateral zones, where TTC[neg] and CD31[neg] co-localized (Fig. 4c, d). Plasma cTNI levels indicated establishment of infarction, but linearity was not statistically significant for the assessed infarction sizes (Fig. 4f). Of note, infarct size

**Fig. 4** Correlation of echocardiography, cardiac troponin I (cTNI) plasma levels, triphenyl tetrazolium chloride negative (TTC[neg]) areas and CD31 negative (CD31[neg]) volumes after 24 h of reperfusion in the same 5 mice. **a** Top panel: Correlation of CD31[neg] volumes and TTC[neg] areas per slice (left) and heart (right) of various infarct sizes (left: $n = 40$ slices from 8 hearts; right: $n = 8$ hearts; r- and p-values depicted). Bottom panel: Associated Bland-Altman analyses of CD31[neg] volumes and TTC[neg] areas per slice (left, −1.08% mean difference) and per heart (right, −0.75% mean difference) of correlations depicted above (dashed black lines mark 95% confidence intervals, solid black line marks zero bias, red dashed line marks mean difference). **b** Inter-operator CD31[neg] volume quantification variances. (Top) Absolute quantified volume of 3 users in 5 randomly selected slices of different hearts subject to 45 min of ischemia and 24 h of reperfusion. (Bottom) The quantification results per user relative to the average (100%, dashed line) quantified volume (in comparison to left, slice #4 was excluded as no CD31[neg] volume was measurable). **c** Original micrographs of echocardiographic imaging. Dashed lines indicate left ventricular lumen. White stars indicate zones of abnormal left ventricular wall movement. **d** Same I/R injury heart as in **c**: exemplary TTC staining and subsequent light sheet fluorescence microscopy (LSFM)-based 3D reconstruction of the same slice. **e** Correlation of ejection fraction (EF) reduction with TTC[neg] areas and CD31[neg] volumes, respectively, per heart ($n = 5$ hearts; r- and p-values depicted). **f** Left: raw data used for correlation of EF reduction, TTC[neg] and CD31[neg] with plasma cTNI levels (value for each animal given, $n = 5$). Right: correlation of absolute cTNI levels with EF reduction, TTC[neg] and CD31[neg]. Scale bar values in µm. Source data are provided as a Source Data file

determination by troponin assessment has been rather difficult in both, clinical and experimental environments[14,25]. Finally, a comparison of 3 operators regarding CD31[neg] volume quantification yielded < 15% difference of average values (Fig. 4b). Interestingly, we observed a wide spread of infarct-body sizes relative to the AAR (Supplementary Figure 5). Notably, several zones low on TTC (TTC[low]), which might be considered I/R injury, were not co-localized with CD31 negativity (Fig. 4 and Supplementary Figures 4 and 5).

**3D vascular ultrastructure during I/R injury recovery**. The CD31 label allowed to reconstruct the 3D appearance of the cardiac vasculature in detail. We observed a highly organized capillary network (Fig. 5a). 2D quantification of vessel densities revealed no significantly different values within the heart muscle, but high variances concerning the field of view in the same heart region (Supplementary Figure 6a). We observed a variability in directionality of vessels, which seemed to be linked to localization within the muscle (Supplementary Figure 6b). In addition, 3D analysis of the left ventricular wall displayed continuous tracts of capillaries surfacing multiple times in one image, which might explain counting errors in 2D immuno-histology (Supplementary Figure 6c). We next employed a filament-tracing pipeline[18] to quantify length and complexity of the vasculature in 3D. In the left ventricular wall we found a vessel length density (VLD) of 2,946 mm per mm3 and an interbranch distance (IBD) of 67 µm (median; $n = 3$) reflecting a > 2× higher VLD compared to brain (Fig. 5d, e and Supplementary Figure 6d)[18]. Hence, the blood vessel architecture is a characteristic hallmark of intact cardiac tissue.

Following I/R injury, the heart muscle undergoes a complex and dynamic healing process[26]. To investigate whether such structural alterations were detectable by changes in the expression pattern of CD31, we investigated hearts 24 h and 5 days following I/R by LSFM (Fig. 5b, c). At day 5 post-I/R, we only found small CD31[neg] areas throughout the damaged heart when compared with cardiac samples after 24 h. With special regard to injured regions, we observed shorter, less oriented and more branched vessels within the damaged area at day 5 post-I/R. We termed this zone curly (CD31[curly]) to describe its distinctive morphological features. Compared to healthy myocardium, filament tracing in curly areas revealed a trend for lower VLD (1,771 mm/mm$^3$, median, $n = 3$) and smaller IBD (51 µm, $n = 3$, median) values, which principally confirms the initial observation of a shorter and more complex vascular network at sites of injury (Fig. 5d, e).

To better localize the I/R injury-zones, CD31[neg] areas were overlaid to a well-known 17-segment heart muscle scheme[27] (Fig. 6a and Supplementary Figure 7). Apical and lateral heart-segments were primarily affected (Fig. 6b). After 5d of reperfusion, the detectable I/R injury was largely resolved, with

only residual peri-apical CD31[neg] segments remaining. The analysis also suggested that CD31[curly] zones on day 5 co-localized with previous CD31[neg] I/R regions. Hence, curly zones might serve as a proxy to delineate previous areas of I/R injury, even days after their revascularization (Fig. 5b, c and 6b).

**3D mapping of immune cells to zones of infarction**. To investigate the impact of cardiac tissue injury on inflammatory response mechanisms, we next assessed immune cell infiltration of I/R-affected zones. To this end we localized and quantified neutrophils and macrophages in the infarcted heart after 24 h and 5 days of reperfusion. In line with previous studies and flow cytometric analyses of injured hearts, we found that Ly-6G positive (Ly-6G[pos]) neutrophils were most prominent after 24 h (Fig. 7a–c and Supplementary Movies 5–8). Intriguingly, the localization of neutrophils matched with CD31[neg] areas, but did partly not overlap with TTC[neg] zones (Supplementary Figure 8). Neutrophil numbers strongly decreased after 5 days, whereas F4/80 positive (F4/80[pos]) macrophages increased. At 24 h we found neutrophils to be highly enriched within the AAR and at the rim of infarct bodies (Supplementary Movie 5) indicating neutrophil presence at the border of infarction rather than within the damaged tissue itself (Fig. 7d and Supplementary Movies 7 and 8). Interestingly, at d5, the majority of neutrophils was no longer associated with the border of the residual infarct-body but localized in the CD31[curly] volume (Fig. 7d, $n = 5$, and Supplementary Movie 6). At this time, we could also detect F4/80[pos] macrophages throughout this area (Fig. 7e, $n = 5$). Such a timing of arrival might reflect the ongoing tissue repair mediated by macrophages[26].

## Discussion

We developed BALANCE for the fast 3D characterization of damage- and response-mechanisms in whole heart specimens. The ability to reconstruct the 3D shape of affected zones might require to re-define the current 2D-concept of myocardial infarction parameters into Volumes At Risk (VAR)/infarct-bodies to give credit to their real 3D nature.

The established protocol combines the use of non-toxic reagents, short handling and protocol times, less complexity and superior signal homogenization, required for myocardial I/R analysis and not available from other published clearing methods (Supplementary Table 2). In contrast to iDISCO clearing, ethanol was used for sample dehydration to avoid high methanol toxicity and reduce possible incompatibilities with following histological antibody stainings[10]. Also, long clearing times, e.g. in CUBIC[28] or passive CLARITY[29], were circumvented by the use of non-toxic ECi. BALANCE uses a minimum of hands-on time and procedure steps to circumvent initial adaption difficulties as

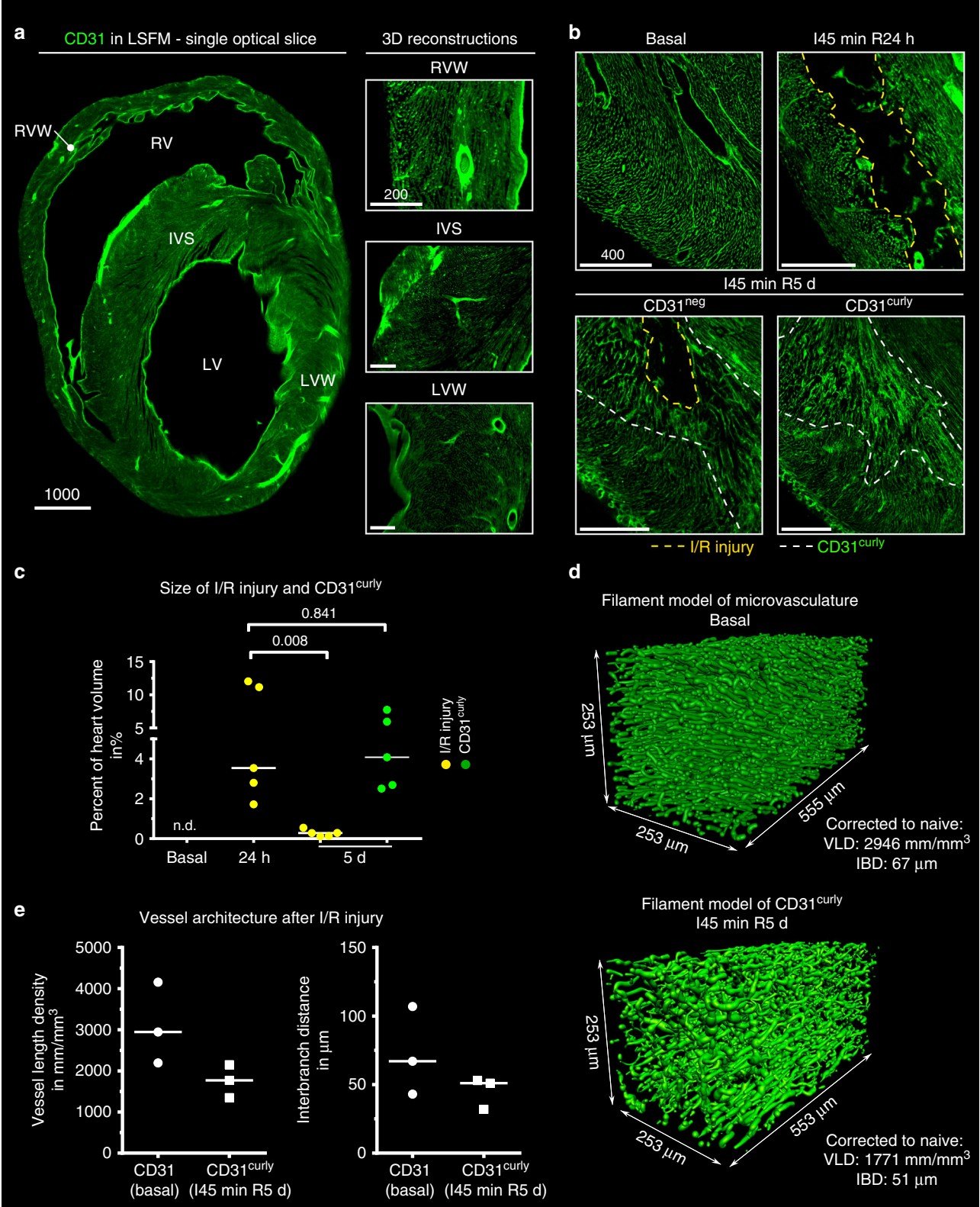

**Fig. 5** 4D analysis of vascular meshwork in basal and infarcted hearts. **a** Left: single optical slice through a cleared murine heart depicting CD31 (green). Right: 3D reconstructions of enlargements, visualizing the microvasculature and its respective directionality. Magnification: ×10. **b** Maximum intensity projections (MIPs) comparing CD31 staining in basal, 24 h and 5 d murine hearts. I/R injury is highlighted with yellow dashed line, CD31 curly (CD31$^{curly}$) region with white dashed lines. Magnification: ×6.4 (basal), 3.2 × (24 h), 4 × (5 d). **c** Quantification of I/R injury and CD31$^{curly}$ volume normalized to heart muscle volume (including appendages and right ventricle (RV)) over time ($n = 5$ hearts; median; two-sided Mann–Whitney test, p-value depicted). **d** Filament model of CD31 baseline expression (top) and CD31$^{curly}$ region (bottom). Magnification: 6.4 × . **e** Vessel length density (VLD, left) and interbranch distance (IBD, right) in baseline and CD31$^{curly}$ regions (ROIs from 3 hearts; median). Scale bar values in µm. Source data are provided as a Source Data file

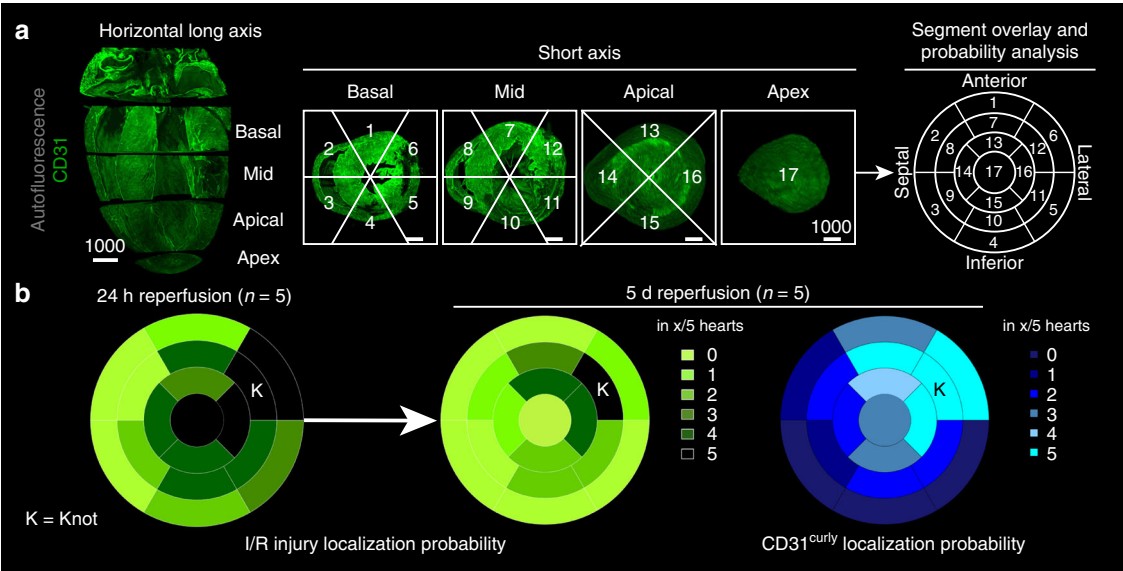

**Fig. 6** Quantifying I/R injury, AAR and CD31[curly] localization in mice with the left ventricular 17-segment model. **a** Left: 3D model of inferior heart half, cut along the horizontal long axis, based on CD31 labelling (green). Four slices are defined: Apex (1), not containing left ventricle (LV) lumen, Apical, Mid and Basal, which are equally spaced with LV lumen. Middle: These slices, viewed from the apex perspective, can be further segmented. The segments do not include the right ventricle. Right: Overlay of the segments viewed from the apex gives a backbone for probability heat-maps. **b** The localization probability of I/R injury after 24 h (left) and 5 d (middle) together with CD31[curly] (right) given in the total amount of hearts affected, represented in heat-maps. K = position of knot in mid anterolateral (segment 12). Scale bar values in µm. Source data are provided as a Source Data file

compared to SWITCH[30]. Most importantly, it homogenizes tissue autofluorescence throughout highly autofluorescent organs such as the heart, which is not the case in the original ECi protocol[9] (Supplementary Table 1 and 2). With the advent of robotics and artificial intelligence (AI) to automate clearing procedures and data analyses[9], the usability and reliability of clearing methods and staining results will be further improved.

Regarding time and expense of experiments, we compared the conventionally applied methods in the field of I/R injury in the murine heart with our current LSFM-based approach (Supplementary Table 3). The hands-on time of our proposed LSFM procedure is comparable to each of the classical methods[14]. However, none of the currently available approaches, including TTC staining, histology, immuno-histology and flow cytometry provides LSFM-comparable data. LSFM combines spatially resolved I/R injury parameter assessment with cellular quantification and 3D localization in the same heart. Furthermore, LSFM-based analysis offers, inherent from its 3D nature, the possibility to reconstruct the fine ultrastructure of the inspected feature such as the blood capillary network. We therefore believe that with our proposed method, the robustness over experiments can be increased. At the same time, combined experimental times needed to carry out all other individual methods in different animals can be significantly reduced by our multiplexing approach. In the near future, computer-based algorithms will further reduce analysis time.

Since our technique can be translated to any I/R injury model, this work opens multiple possibilities for identifying new therapeutic targets. Of note, clinical data on cardioprotection have been largely disappointing[31] and morbidity and mortality for patients with an acute myocardial infarction remain considerably high[32]. Novel approaches are therefore desired to improve the outcome of patients. The underlying pathomechanisms in I/R injury are very complex and include e.g. the interaction of heart cells (cardiomyocytes, endothelium, fibroblasts) with platelets and immune cells finally producing heart cell death. It has recently been concluded that an ideal drug/approach for cardioprotection would target the cardiac vasculature, immune cells and cardiomyocytes[14,31]. Several promising strategies have been forwarded recently including beta-blocking metoprolol[33–35] and immune modulatory drugs[36,37]. A common feature of these approaches is that they target not a single but multiple pathways involved in I/R injury. Metoprolol, e.g., interacts with beta-receptors on cardiomyocytes to reduce myocardial energy consumption while also inhibiting neutrophil-platelet interaction involved in microvascular obstruction[33–35]. With our proposed workflow, we provide a tool for the analysis of conventional I/R injury in conjunction with the characterization of the vascular injury several days after the initial I/R event. Another problem of previously forwarded potentially cardioprotective therapies appears to be the lack of rigor in experimental studies[14]. To establish rigor in our model, we performed benchmarking. Hence we have conducted correlation analyses of our proposed analysis tool for the CD31[neg] I/R injury against recently recommended standard approaches for experimental animal studies[14] and show a very good correlation (Fig. 4)[7,8,14,23,38]. Of note, instead of using CD31 antibody labelling to stain the vasculature, a lectin-based approach would obtain comparable results, depending on the investigator's choice[39–41].

Remarkably, the vasculature has been largely affected as shown by our study and the vascular injury correlates with tissue damage as assessed by classical methods. It is therefore tempting to speculate that the vasculature could become a primary target for future cardioprotective strategies. Hence, we believe that BALANCE/LSFM might also help to identify additional potential novel therapeutic options for the treatment of I/R injury in the future.

## Methods

**Animals**. All experimental animal procedures were approved by the responsible governmental agency, the Ministry for Environment, Agriculture, Conservation and Consumer Protection of the State of North Rhine-Westphalia (MULNV) and complied with all relevant ethical regulations for animal testing and research. Male C57BL/6JRJ (12 ± 3 weeks of age; Janvier) and Catchup (C57BL/6-Ly6g(tm2621

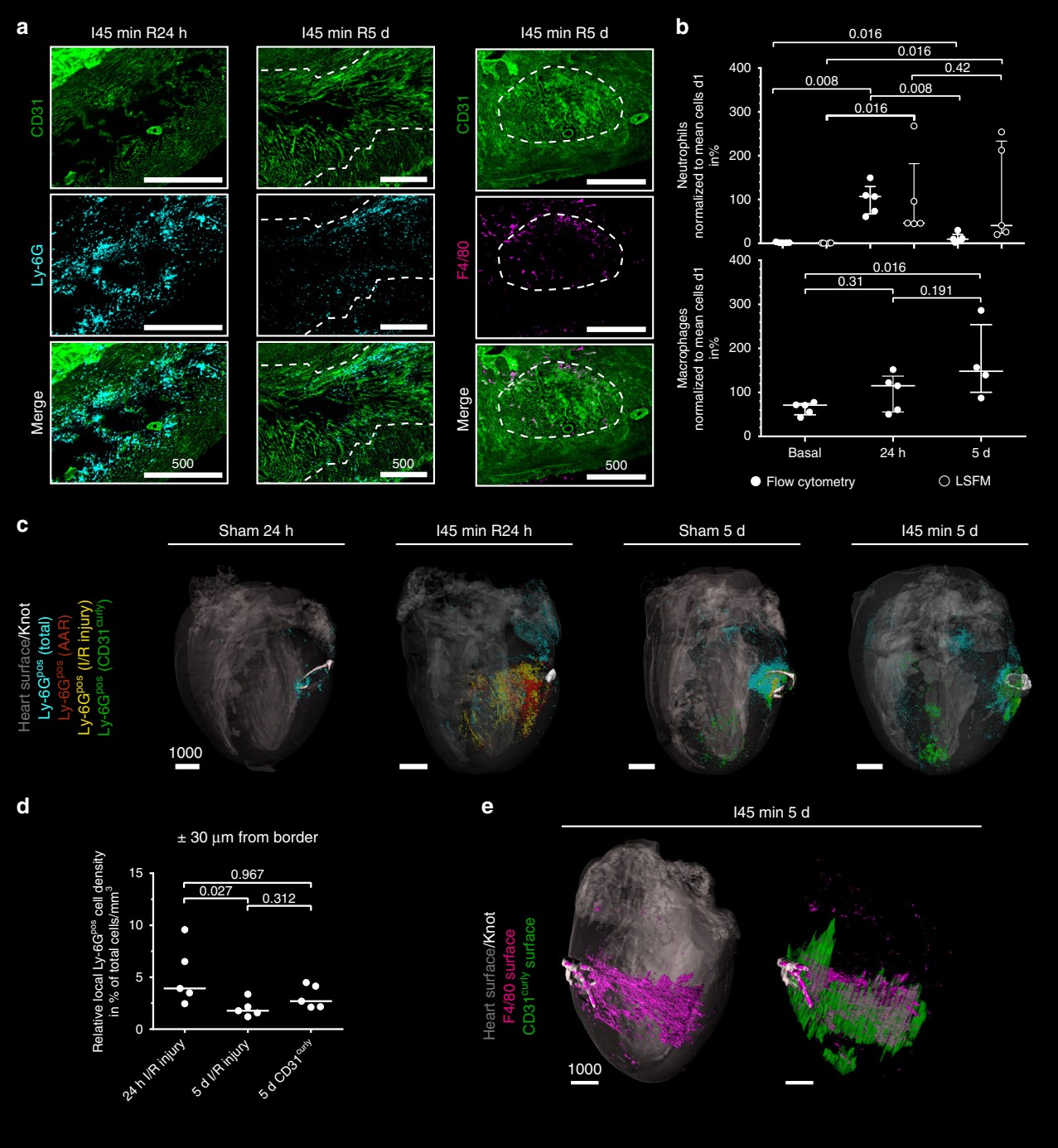

**Fig. 7** Characterization of immune cell infiltrates into I/R injury. **a** Example maximum intensity projections (MIPs) of Ly-6G positive (Ly-6G[pos]) neutrophils after 24 h (left) as well as Ly-6G[pos] neutrophils and F4/80 positive (F4/80[pos]) macrophages after 5 d (middle; right). Depicted are CD31 (green, top row) and Ly-6G (turquoise) or F4/80 (purple) signals (middle row) together with a merge (bottom). Dashed white line marks the border of the CD31[curly] region. Magnification: ×3.2 (24 h), 4 × (5 days). **b** Immune cell quantities (top neutrophils, bottom macrophages) determined by flow cytometry (white, filled dots) or based on LSFM (white, empty dots) analysis after 24 h and 5 d of reperfusion, as well as under baseline condition (n = 4–5; median ± interquartile range; Kruskal–Wallis test with Dunn's correction). **c** Representative 3D visualization of Ly-6G[pos] cell distribution across whole murine hearts. Heart structure (grey,) and knot (white) are shown for orientation. Ly-6G[pos] cell populations are colored in respect to their localization in either AAR (red), I/R injury (yellow), CD31[curly] (green) or outside the above-mentioned structures (turquoise). **d** Relative cell density ± 30 μm from the respective volume border (n = 5; median; Kruskal–Wallis test with Dunn's correction). **e** 3D visualization of F4/80[pos] signals depicted as a surface rendering (purple) in relation to heart surface (left; heart volume in grey) or CD31[curly] surface (right; CD31[curly] in green) after 5 d of reperfusion. The knot (white) is visualized for orientation. Scale bar values in μm. This finding was verified in 5 mice. Source data are provided as a Source Data file

(Cre-tdTomato)Arte); 12 ± 3 weeks of age)[42] mice were held at the local animal house on 12 h/12 h day and night cycle with water and food *ad libitum*.

**Human studies**. All human studies were approved by the local ethics committee of the Medical Faculty, University of Duisburg-Essen, Germany (Ethics Approval Nr. 18–8527-BO).The whole procedure was in accordance to the World Medical Association Declaration of Helsinki and conducted in accordance with the relevant ethical regulations. Written informed consent was obtained from all participants. The patients underwent coronary bypass surgery, during which human left atrial appendage (LAA) biopsies were routinely excised. This presented no further harm for the patient.

**Ischemia/Reperfusion (I/R) heart injury**. Male C57BL/6JRJ and Catchup mice (12 ± 3 weeks of age) were subjected to a published myocardial I/R in vivo protocol[38]. Briefly, mice were anesthetized by intraperitoneal (i.p.) injection of ketamine (100 mg/kg, CatNo. 9089.01.00, bela-pharma) and xylazin (Rompun 10 mg/kg, CatNo. 6324464.00.00, Ceva Tiergesundheit). They were orally intubated and ventilated throughout the operation procedure, with 0.8 l/min air and 0.2 l/min $O_2$ at a tidal volume of 250 µl/stroke and a breathing frequency of 140 strokes/min. Anesthesia was maintained during the operation by supplementing 2% isoflurane (Forene, CatNo. 2594.00.00, abbvie). The chest was opened through a left lateral thoracotomy and the left coronary artery (LCA) was ligated. After 45 min of ischemia, reperfusion was allowed for indicated time points. As previously described, different infarct sizes may be created by this method due to differences in mouse physiology[43,44]. Mice were treated with 0.1 mg/kg buprenorphine (Temgesic, CatNo. 997.00.00, Indivior) subcutaneously every 8 h after operation for a total of 72 h. Mice received 1000 IE heparin (CatNo. 27586.00.00, LEO Pharma) i.p. 10 min before the end of the experimental protocol and were killed by cervical dislocation.

For light sheet experiments, mice received intravenous (i.v.) tail vein injections 10 min before sacrifice with 5 µg of each antibody in PBS in a total volume of 150 µl per animal of anti-mouse CD31 (clone: Mec13.3; purified: CatNo. 553369, BD; conjugated with AlexaFluor (AF) 647: CatNo. 102516 BioLegend), Ly-6G (clone: 1A8; purified: CatNo. 127602, BioLegend; conjugated with AF647: CatNo. 127610, BioLegend) and/or F4/80 antibody (clone: BM8; purified: CatNo. 123102, BioLegend; conjugated with AF594: CatNo. 123140, BioLegend). In other experiments, respective IgG control antibodies were used (rat IgG2a-AF647, CatNo. 400526, BioLegend; rat IgG2a-AF594, CatNo. 400555, BioLegend) (Supplementary Figure 9). Purified antibodies were conjugated with AF790 using an IgG coupling kit (CatNo. A20189, Thermo Fisher). After sacrifice, mice were perfused blood-free with phosphate-buffered saline (PBS), hearts were excised and subjected to following tissue processing.

To visualize the area at risk (AAR) for light sheet experiments, the heart was placed in ice-cold PBS after perfusion, the aorta was cannulated, the LCA re-occluded and 2% gelatin / 0.2% fluorescein isothyocyanate (FITC)-albumin in PBS (gelatin CatNo. G9391, Sigma Aldrich; FITC-albumin, CatNo. A9771, Sigma Aldrich) was injected via the aorta. After 10 min incubation in ice-cold PBS, hearts were subjected to tissue processing.

For hearts to be analyzed by both light sheet fluorescence microscopy (LSFM) and triphenyl tetrazolium chloride (TTC) staining, freshly excised hearts were cut transversely along the longitudinal axis in 2 mm slices and stained for 5 min at 37 °C with a 1% (w/v) TTC in 0.0774 M $Na_2HPO_4$ / 0.0226 M $NaH_2PO_4$ in dd$H_2O$ solution ($Na_2HPO_4$, CatNo. 4984.2, Roth; $NaH_2PO_4$, CatNo. S5011, Sigma Aldrich). Subsequently, macroscopic images of the TTC-stained slices were taken with a M80 microscope with an IC80 HD camera at 1.6× – ×2.5 magnifications (both Leica) before being subjected to tissue processing. Computer-assisted planimetry was performed in a double blinded fashion using ImageJ software[45].

Mouse plasma was prepared from whole blood taken from inferior vena cava shortly before heart extraction, mixed with 25 IE heparin (CatNo. 27586.00.00, LEO Pharma) and centrifuged 10 min at 3000 × *g*, 4 °C. Supernatant (plasma) was collected, snap-frozen in liquid $N_2$ and stored at −80 °C until analysis.

**Quantification of I/R injury size using TTC staining**. I/R injury size was quantified as described before[38]. Briefly, after PBS perfusion and extraction, the LCA was re-occluded and 1 ml 1% Evans blue dye (CatNo E2129, Sigma Aldrich) in 0.9% NaCl was injected via the aorta to visualize the AAR. The heart is wrapped in clear food wrap and stored at −20 °C for 1 h. The heart is then cut in 2 mm slices orthogonal to the long axis and incubated in a 1% (w/v) TTC solution as described above. Areas of TTC^neg, AAR and non-ischemic myocardium were measured using ImageJ[45] by two blinded, independent operators. I/R injury size was expressed as percentage of AAR.

**BALANCE protocol for tissue clearing**. Perfused hearts were immersed in 4% paraformaldehyde (w/v, PFA, CatNo. 10195, Morphisto) in PBS for chemical fixation while standing for 4 h at 4 °C in 15 ml tubes. For all following steps, hearts were kept at 4 °C, always agitated, transferred only to pre-cooled solutions in 15 ml tubes and kept protected from light. Following fixation, hearts were dehydrated in an ascending ethanol series in dd$H_2O$ (v/v) of 50%, 70 and 100% (CatNo. 9065.2,

Roth) for at least 4 h while shaking. Samples can also be stored longer at each step without loss of staining quality.

Subsequently, samples were bleached for 4 h in freshly prepared 5% (v/v) hydrogen peroxide (CatNo. 349887; Sigma Aldrich) and 5% (v/v) dimethyl sulfoxide (CatNo. 4720.3, Roth) in 100% ethanol. Please beware of building pressure during this step. Bleaching was carried out to enhance sample clarity, enabling homogenous imaging in lower wavelength channels while preserving artificial fluorophore fluorescence (Fig. 1). After a washing step of at least 4 h in 100% ethanol, samples were warmed to room temperature (RT) for 5 min before transfer into 7 ml pure (99%) ethyl cinnamate (ECi, CatNo. 112372, Sigma Aldrich) in a glass vial for at least 4 h prior to imaging. Samples were kept at RT in the dark until and after imaging.

**Additional bleaching protocols tested**. Further tested bleaching protocols were performed after fixation and washing. Therefore, perfused hearts were immersed in 4% PFA in PBS for chemical fixation while standing for 4 h at 4 °C in 15 ml tubes, as mentioned above. Hearts were washed in 5 ml PBS + 0.01% sodium azide (w/v, Cat. No. K305.1, Roth) over night at room temperature (RT), shaking. Afterwards, hearts were washed again another change of 5 ml PBS + 0.01% sodium azide for 1 h at RT, shaking.

Sudan Black bleaching was performed according to the original publication[15]. After fixation and washing, hearts were incubated in 0.5% (w/v) Sudan Black B (w/v, Cat. No. 199664, Sigma-Aldrich) in 70% ethanol in dd$H_2O$ for 3 h at RT, shaking. After washing 3x in 5 ml PBS + 0.01% sodium azide (w/v, Cat. No. K305.1, Roth) for 30 min at RT, shaking, hearts were dehydrated in ascending ethanol series in dd$H_2O$ (v/v) of 50%, 70 and 100% for at least 4 h at 4 °C while shaking. Afterwards, hearts were warmed to RT for 5 min before transfer into 7 ml pure (99%) ECi in a glass vial for at least 4 h prior to imaging. Samples were kept at RT in the dark until and after imaging.

Bleaching using heme elution was accomplished as published beforehand[15]. After fixation and washing, hearts were immersed in 50% CUBIC-1 reagent (v/v) in dH$_2$O for 3 h at 37 °C, shaking. CUBIC-1 reagent was prepared according to published protocols[28] and was comprised of 25% Urea (w/w, Cat. No. U5378, Sigma-Aldrich), 25% Quadrol (w/w, Cat. No. 122262, Sigma-Aldrich) and 15% Triton X-100 (w/w, Cat. No. X-100, Sigma-Aldrich) in dH$_2$O. Afterwards, hearts were incubated in pure CUBIC-1 reagent over night at 37 °C, shaking and further in another change of CUBIC-1 reagent for 2 d at 37 °C, shaking. Following, hearts were dehydrated in ascending ethanol series in dd$H_2O$ (v/v) of 50%, 70 and 100% for at least 4 h at 4 °C while shaking. Afterwards, hearts were warmed to RT for 5 min before transfer into 7 ml pure (99%) ECi in a glass vial for at least 4 h prior to imaging. Samples were kept at RT in the dark until and after imaging.

**CUBIC clearing protocol**. CUBIC clearing protocol was performed as previously described[28]. All following incubation steps were carried out in the dark. After perfusion, hearts were immersed in 4% PFA in PBS for chemical fixation while standing for 4 h at 4 °C in 15 ml tubes, as mentioned above. Hearts were washed in 5 ml PBS + 0.01% sodium azide (w/v, Cat. No. K305.1, Roth) over night at room temperature (RT), shaking. Afterwards, hearts were washed again another change of 5 ml PBS + 0.01% sodium azide for 1 h at RT, shaking. Afterwards, hearts were immersed in 50% CUBIC-1 reagent (v/v) in dH$_2$O for 3 h at 37 °C, shaking. CUBIC-1 reagent was prepared according to published protocols[28] and was comprised of 25% Urea (w/w, Cat. No. U5378, Sigma-Aldrich), 25% Quadrol (w/w, Cat. No. 122262, Sigma-Aldrich) and 15% Triton X-100 (w/w, Cat. No. X-100, Sigma-Aldrich) in dH$_2$O. Following, hearts were incubated in pure CUBIC-1 reagent over night at 37 °C, shaking and further in other changes of CUBIC-1 reagent for 2 d each for a total of 4 d at 37 °C, shaking. Hearts were then washed in three changes of PBS + 0.01% sodium azide for 2 h each at RT, shaking. Next, hearts were immersed in 50% CUBIC-2 reagent (v/v) in PBS for 6 h at 37 °C, shaking. CUBIC-2 reagent was comprised of 25% Urea (w/w), 50% Sucrose (w/w, Cat. No. S7903, Sigma-Aldrich) and 10% Triethanolamine (w/w, Cat. No. 90279, Sigma-Aldrich) in dH$_2$O following establish protocols[28]. Afterwards, hearts were incubated in pure CUBIC-2 reagent over night at 37 °C, shaking and another change of CUBIC-2 reagent for 24 h at 37 °C, shaking. Hearts were imaged in a 50%/50% silicon oil (Cat. No. 175633, Sigma-Aldrich) and mineral oil (Cat. No. M5904, Sigma-Aldrich) mixture.

**Light sheet fluorescence microscopy and image processing**. Samples were imaged using an Ultramicroscope II and ImSpector software (both LaVision BioTec). The microscope is based on a MVX10 zoom body (Olympus) with a 2x objective and equipped with a Neo sCMOS camera (Andor). For image acquisition, cleared samples were immersed in ECi in a quartz cuvette and excited with light sheets of different wavelengths (488, 561, 639 and 785 nm). Following band-pass emission filters (mean nm / spread) were used, depending on the excited fluor-ophores: 525/50 for FITC; 595/40 for AF594 or autofluorescence; 680/30 for AF647 and 835/70 for AF790. For quantitative imaging, whole hearts were imaged along the longitudinal axis. For image acquisition, hearts were trapped, with the apex and the aorta horizontally aligned, in a commercially available sample holder (LaVision BioTec). To avoid damage or deformation of the sample, ECi-cleared 1% phytagel/ $H_2O$ (CatNo: P8169–100G, Sigma Aldrich) blocks were used as buffers

between tissue and plastic holder. Hearts were also rotated along the longitudinal axis, so that the knot, which remains in situ, faced downwards. This reduced blockage of excitation or emission light to a minimum. Whole-heart data sets were obtained with ×2 total magnification (pixel size of 3.25 μm / pixel x,y, lateral resolution: 6.5 μm in $x$ and $y$) with 10 μm z spacing between optical planes. Since the camera's chip has 2560 × 2160 pixels, the field of view with this magnification amounts to 8.32 × 7.02 mm. It was thereby possible to image a whole mouse heart without multi-positioning, thus avoiding stitching algorithms while simultaneously reducing acquisition times. Sheet width was set to 4200 and numeric aperture to 0.148, resulting in an approximate light sheet thickness of 4 μm in the horizontal focus. Illumination time was 350 ms, with enabled 8x dynamic focus in both left and right laser lines. Total acquisition time of one plane was thus, theoretically, 5.6 s. The vertical thickness of the hearts was generally smaller than 6 mm. This is the limiting factor due to the objective's working distance. Assuming a z stack as deep as 6 mm, this would have resulted in acquisition of 600 images × 4 channels = 2400 images, which would theoretically have taken 3.73 h. However, in our practical experience one murine heart takes 4–6 h of acquisition time. For regions of interest (ROIs), magnifications are indicated in the figure legends.

See https://www.lavisionbiotec.com/products/UltraMicroscope/specification.html for zoom factors, corresponding numerical apertures and resolutions. To avoid photobleaching during imaging, the longest wavelength was imaged first.

16 bit OME.TIF stacks were converted (ImarisFileConverterx64, Version 9.2.0, BitPlane) into Imaris files (.ims). 3D reconstruction and subsequent analysis was done using Imaris software (BitPlane). Heart surface (25 μm grain size) and volume was determined using surface creation algorithms. All surface tracings (AAR, CD31[neg], CD31[curly], vessels) are based on the contour tracing tool and are carried out manually/semi-automatically. For surface creation, each (vessels), every 5[th] (CD31 tracings) or 10[th] (AAR) image was traced. Surfaces were created with maximum resolution (2160 × 2560 pixel) and preserved features, to ensure matching of the traced lines with the surface border. Discrimination of arteries and veins was done by their respective location within the heart muscle and the signal intensity of the CD31 staining (CD31[high] for arteries, CD31[low] for veins).

For 2D vessel counting, 8 slices of the right ventricular wall (RVW), the intraventricular septum (IVS) and the left ventricular wall (LVW) ($n = 3$ mice) were analyzed. For this, capillaries in 8 fields of view of 251.49 × 251.49 μm size (not shrinkage corrected) were manually counted and extrapolated to capillaries per mm².

Vessel quantification was performed as previously described[18]. Image data were first preprocessed, as a necessary step to reduce noise and improve vessel contrast, before applying a filament tracer model for vessel quantification in the 3D software package Imaris (BitPlane).

Briefly, image stacks were first preprocessed in the open source software ImageJ[45] by applying a Gaussian smoothing, where a Gaussian sigma of 2 μm was chosen to be less than half the diameter of the smallest vessels. This was followed by a rolling ball background subtraction. In order not to affect the intensity distribution within the vessels themselves, a rolling ball radius of 20 μm was chosen, which was more than twice the diameter of the largest vessel. Any resulting edge artifacts from background variations could be removed during a later thresholding step in Imaris. The contrast of vessels was further enhanced using a python script (vmtkimagevesselenhancement) from the open source Vascular Modelling Toolkit project (VMTK: www.vmtk.org), which performed a multiscale Hessian-based Frangi vesselness filter[46]. The feature-based enhancement filter of Frangi is now a well-established tool for vessel segmentation[47]. In addition to the improved vessel contrast and non-vessel suppression, the Frangi filter also improved the continuity of the vessels by smoothing out inconsistencies in the vessel labeling along the length of the vessels. These preprocessing steps were found to be necessary for the fidelity of tracing smaller vessels when applying the filament-tracing algorithm in Imaris. In particular since, for the tracing of vessels, it was necessary to apply the 'with loops' filament-tracing method, which uses an initial simple thresholding for segmenting the vessels. Without smoothing, background subtraction and contrast enhancement, small vessels that were less intense were either inconsistently segmented, due to variations in the background signal, not distinctly segmented when close to larger much brighter vessels, or were not segmented as an integral vessel, but were broken up due to inconsistencies in the labelling efficiency. It should also be noted that one known artifact of the Frangi filter is that it can lead to discontinuities at branching points. In particular, for branching that occurs in a direction orthogonal to the main branch. Although we observed a reduction in the vessel intensity at these points, this effect occurred at a range very close to the branch point, and for our data did not result in any discontinuities in the Imaris tracing. Furthermore, we also observed that the Frangi vessel filter can affect the relative intensities of the vessel branches. Indeed, the intensity of smaller vessels was seen to be enhanced relative to larger vessels. In the Imaris filament tracer, this could influence the evaluation of their relative diameters, due to the use of the simple intensity threshold method for segmentation. Indeed, in general, the application of a simple thresholding segmentation resulted in a relative mismatch in the filament diameters between small low intensity vessels and the larger brighter vessels, and also resulted in a mismatch for the filament path, calculated from the center lines of the segmented vessels. Therefore, following the construction of the initial filament model, this mismatch was corrected in Imaris with two further processing steps. Firstly, a 're-alignment' of the filament model was performed against the channel

corresponding to the Gaussian smoothed and background subtracted vessel data. The same channel was then used to adjust the filament diameters, using an approach based on the relative contrast change across the diameter of the vessels. From the resulting filament model, parameters for the vascular length and branching points could be extracted.

For quantification of CD31 signal intensities over different specimens, single optical planes and MIPs of whole hearts were manually analyzed.

In order to determine the tissue shrinkage occurring during the BALANCE protocol we measured heart slice thickness after heart extraction (naïve, 1.9 mm ± 0.17; mean ± s.d.) and shortly before light sheet imaging (cleared, 1.555 mm ± 0.12; mean ± s.d.) in 3 mice. The slice height was reduced to 82% ± 2% (mean ± s.d.) of their original size. The factor used to correct all displayed volumes throughout the manuscript was therefore 1.22 (1 divided by 0.82) for every spatial dimension. Please note that displayed scale bars are not corrected, since they relate to the real size of the samples imaged within ECi.

Immune cell counts were determined using the spot detection algorithm with an assumed diameter of neutrophils of 7 μm (14 μm z). After assessment of signal distributions in non-stained, basal, I45min R24h and I45min R5d hearts, a threshold was applied for all hearts (4500–35000 grey values, Supplementary Figure 9a and b). Localization of detected spots respective to surface borders was measured using the distance transformation extension on desired surfaces. As a result, spot intensity values are replaced by the minimum distance to the border of the transformed surface. The sum of all detected spots within ± 30 μm of the surface borders of infarct bodies and CD31[curly] regions were normalized to the respective volume and the total amount of cells present in the heart. We used a transgenic mouse line with endogenously labeled neutrophils (Catchup mice[42]) to show that all neutrophils invading the myocardium are labeled using our i.v.-mediated staining approach (Supplementary Figure 2).

For the 17-segment model, clipping planes were introduced in the required locations of the left ventricle (see Cerqueira et al[27]), images of 3D content per slice were taken and overlaid with a segment grid in Illustrator CC (Adobe). If a segment showed I/R injury/AAR/CD31[curly], it was digitally rated as positive. The sum of the counts per segment over all hearts was depicted as heat maps, generated using R software[48].

Due to the diffuse staining pattern of the F4/80 AF790 antibody, spot detection was not feasible, and we rendered the volume of the immune cell localization instead.

Enlargements and other displayed content was processed using ImageJ software[45].

**Echocardiographic assessment**. Mouse echocardiography was conducted using Visualsonics Vevo 2100 Imaging system. After initial anesthesia with 4% isoflurane in 1 L/min $O_2$, mice were placed on a heated plate and anesthesia was maintained with 1.5% isoflurane. A rectal probe was introduced and heart rate, respiratory rate and body temperature were monitored continuously. After hair removal, images of long and short parasternal axis were acquired in M-mode and B-mode. Ejection fraction was calculated using Simpson's method[49].

**Cardiac troponin I ELISA**. Cardiac troponin I levels were determined from heparin plasma using ultra-sensitive mouse cardiac troponin-I ELISA (CatNo CTNI-1-US, Life Diagnostics) following manufacturer's instructions. To obtain quantifiable results, samples were diluted 1:100 in diluent YD25–1, according to supplier's suggestions. Absorbance values were recorded using a FLUOStar Omega (BMG Labtech).

**Flow cytometry procedure and analysis**. PBS-perfused hearts of mice after 24 h and 5 d of reperfusion as well as under basal conditions, were mechanically minced and incubated in an enzyme solution comprised of 450 U/ml collagenase I (CatNo C0130, Sigma Aldrich), 125 U/ml collagenase XI (CatNo C7657, Sigma Aldrich), 60 U/ml hyaluronidase (CatNo H3506, Sigma Aldrich), 20 mM HEPES (CatNo H3375, Sigma Aldrich), 60 U/ml DNase (CatNo D5319, Sigma Aldrich) in PBS for 40 min in a ThermoMixer C (Eppendorf) at 37 °C and 300 r.p.m. After enzymatic digestion, the solution was filtered through a 40 μm filter and flow-through was centrifuged for 5 min at 4 °C and 350 × $g$. Supernatant was discarded and cell pellet resuspended in 1.5 ml PBS. 100 μl of this solution per sample was used for staining, while one solution was chosen for fluorescence minus one (FMO) controls. Blocking of FC-receptors was done by adding 2 μl of TruStain fcX (CatNo 101319, BioLegend) and incubating for 20 min on ice in the dark. Samples were washed by adding 100 μl PBS, centrifuging for 5 min at 4 °C and 350 × $g$, discarding supernatant, and resuspending the cell pellet in 50 μl PBS. Samples were stained by adding 50 μl PBS containing antibodies CD45-AF700 (CatNo 103127, BioLegend), Ly-6G-PerCP/Cy5.5 (CatNo 127615, BioLegend), CD11b-BV605 (CatNo 101257, BioLegend) and F4/80-BV421 (CatNo 123137, BioLegend), all at a dilution of 1:200. Incubation was done for 30 min at RT in the dark, with simultaneous dead and alive staining by addition of Zombie NIR dye (1:2000, CatNo 423105, BioLegend). After antibody incubation, samples were washed as before, supernatant was discarded and cell pellet was resuspended in 150 μl FACS-Buffer, comprised of 1% (v/v) fetal bovine serum (CatNo P30–3302, PAN Biotech) and 0.5% bovine serum albumin (CatNo 8076.3, Roth) in PBS. Data was acquired on a BD FACS

Aria III (BD Biosciences) and analysis was performed using FlowJo software (Ashland, USA). Gating strategy is shown in Supplementary Figure 9c.

**Statistical analysis**. Statistical analysis was performed using GraphPad Prism 6.0 for Windows (GraphPad Software). Data are given as indicated in the figure legends. Student's t-test or Mann–Whitney U test were performed for comparison of two groups. Kruskal–Wallis or one-way / two-way ANOVA (with or without repeated measures) followed by Dunn's multiple comparisons test or Bonferroni's correction were performed for multiple group comparisons, as indicated. Correlation was performed using lineary regression best-fit and calculating Pearson's correlation coefficient r. Comparison of two methods was done by Bland-Altman analysis. P-values are depicted and a value of $p < 0.05$ was considered significant.

## Data availability

The data sets generated and/or analyzed during the current study are available from the corresponding authors upon request. The source data underlying Figs. 1, 4, 5–7, and Supplementary Figure 1, 3, 5 and 6 are provided as a Source Data file.

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

## Acknowledgements
We thank the imaging center Essen (IMCES, https://imces.uk-essen.de) for continuous support of our imaging experiments.

## Author contributions
S.F.M., S.K. developed the BALANCE protocol and performed the LSFM experiments together with J.D. S.F.M., S.K., L.B. analyzed the data, generated all illustrations as well as schematics and prepared the figures. P.S., S.K. and L.M. performed mouse surgeries, T.T.C. planimetry, cardiac troponin I and echocardiographic measurements. A.S. supervised modeling of the vessel system and C.S. translated results in R software. H.L., M.K., J.K., D.R.E., D.M.H., T.R., U.H.C., M.T. and M.G. provided supervision. M.T. and M.G. conceived of the study and wrote the manuscript together with TR, UHC, SFM, SK and LB.

## Additional information

**Competing interests:** M.G. and J.K. received general research funding from LaVision BioTec GmbH. The remaining authors declare no competing interests.

