## [Peer Review File · Nature Communications]

Reviewers' comments:

Reviewer #1 (Remarks to the Author):

The authors have developed a highly novel method to quantify myocardial cell injury and death in a 3-dimensional manner. The authors can also visualize the coronary circulation using the same technique. This technique is convincing and provides some new information. The studies have been performed in a very careful manner and there are no apparent flaws with this work. The authors have compared this method with more traditional TTC and blue dye staining methods and have shown a good correlation. There are some advantages to this new method. However, the time and expense involved appears to be significant and it is not clear if this method can be widely adapted across many laboratories.

The authors suggest that this method will help in the development of novel therapeutics to treat patients that suffer from an acute myocardial infarction and that infarct imaging has been a limitation in the translation of therapies to the clinic. This is simply not the case. The major limitations in the development of therapeutics to treat myocardial infarction in man are: (1) animal models (especially the mouse) do not closely mimic the human condition (2) lack of rigor in preclinical studies (3) myocardial reperfusion injury may not be a significant issue in man, (4) novel cardioprotective targets need to be identified to develop new therapeutics.

It is not clear that this new cardiac imaging modality addresses any of these limitations in the field. At present, the field is at a standstill and there are major questions regarding the utility of studying animal models of myocardial ischemia/reperfusion injury.

This imaging modality is also limited since it requires sacrifice of the animal to perform the imaging. It appears that this method is limited to the murine heart.

Concerns

(1) The authors have generated very small areas of infarction in the mouse heart. It is not clear why the infarct sizes are so small. Studies should be performed to create a variety of sizes of myocardial infarction and evaluate this new imaging method.

(2) The authors should measure circulating and/or myocardial tissue levels of cardiac troponin I in these experiments.

(3) Similarly, the authors should also evaluate cardiac function and structure using echocardiography and correlate with infarct size.

Reviewer #2 (Remarks to the Author):

Nature Commun, Contemporaneous 3D characterization of acute and chronic myocardial I/R injury and response

Merz et al. reported a pipeline for structural analyses of fluorescence-labeled myocardial I/R injury model in the murine hearts. The authors employed a commercial light-sheet imaging system with the bleaching-augmented solvent-based non-toxic clearing (BALANCE) method to characterize and localize the infarcted myocardium in response to I/R injury. The authors demonstrate that BALANCE allows for detailed structure of myocardial infarction on the basis of ethyl-cinnamate as previously reported. While the authors compared BALANCE-LSFM to TTC-histology as a reference, the utility/advantage of their novel clearing method as compared to the current methodology was not articulated. While the authors provide the state-of-the-art imaging to characterize I/R injury patterns and to quantify acute and late vascular I/R damage zones, the imaging data are based on the commercialized LSFM, the clarity methodology was previously reported, and new biological findings are anticipated. Following are some comments to strengthen the manuscript.

Comments

1. The authors used the commercial Ultramicroscope developed by LaVision BioTec to image most samples. The detailed procedures were not articulated in the Method section. Undefined are the total acquisition time, lateral resolution in the cross-sectional image, field of view, light-sheet thickness, and image stitching. These factors are essential to assess the capability of this work. While the authors proposed to image the intact heart along the longitudinal axis, how to precisely position the long-axis along the scanning direction under the microscope was not articulated.

2. In reference to image processing, the authors used Imaris and ImageJ to reconstruct and to render the 3-D hearts. For vessel quantification, the authors applied previous reported method. Uncertain are the potential artifacts introduced by the rolling ball background subtraction and some other processing methods.

3. The authors introduced a new chemical clearing approach or known as BALANCE for LSMF imaging. Unlike their previous publication in which ECI-based method was used (Reference 9), they incorporated a bleaching step to reduce the autofluorescence in this new protocol. However, this would probably quench the fluorescence in the transgenic mice, limiting the potential applications in other genetic models. For this reason, the precise quantification of fluorescence attenuation was not quantified.
4. In reference to Comment #3, the authors used IV injections of fluorescently labeled antibodies. Currently available techniques allow for clearing of heart, followed by labeled-antibody exposure and signal detection. To demonstrate the utility of their technique, the authors may consider the transgenic lines, such as *cmlc-gfp*, to demonstrate the absence of photobleaching and attenuation of the fluorescent signal.
5. Please demonstrate applicability of current method using multi-channel excitation in transgenic mice labeled with 2 or more fluorophores.
6. The advantages of proposed BALANCE method would be strengthened by comparing with other tissue clearing methods, including CLARITY, iDISCO, CUBIC and SWITCH. Please clarify the reason for choosing this specific method in the Discussion section.
7. While the solvent-based clarity method is well-known to cause tissue shrinkage, whether the tissue shrinkage factor based on BALANCE is superior to other solvent-based method such as iDISCO (Cell, 2017) remains unclear.
8. Please provide a detailed protocol for BALANCE for the light-sheet imaging community.

Reviewer #3 (Remarks to the Author):

In this manuscript, Merz and colleagues characterise the 3-dimensional extend of acute and chronic myocardial injury and response. They provide new data on cardiac ischemia and propose a promising new method for further work in that field and tissue clearing. The paper is well written and illustrated with figures of the methodology and results. This work will likely be helpful for future researchers in cardiovascular disease and stroke. The movies are also beautiful and instructive. I have some major and minor concerns about the study:

Major:

1) The primary weaknesses of this study is the 1) lack of hypothesis, and 2) potentially anecdotal nature. I could not find anything in the text about sample size. How many animals were used? On page 12 it said: "8 slices of the right ventricular wall (RVW), the intraventricular septum (IVS) and the left ventricular wall (LVW) of the same heart (n=1) were analyzed. " In figure 2 caption: "volume

normalized to heart muscle volume (including appendages and right ventricle (RV)) over time (n=5; median; two-sided Mann-Whitney test, p-value depicted).” What is n here? Number of animals?

2) Since there is no hypothesis, the objectives should be clearly stated. It seems that part of the objectives of the study primarily to improve the clearing of cardiac tissue, that has “extremely high autofluorescence” in the previously applied method. Nevertheless, if this is the primary goal, why not perform a systematic analysis of the existing clearing methods, rather than ad hoc inventing a new method? Or at least mention other clearing methods and their shortcomings in this context?

Minor:

-Regarding the bleaching with peroxide, it is an effective approach, but also very harsh for the tissue and may damage antigens. I am missing the data or at least the argumentation for selecting peroxide-bleaching compared to other methods such as heme elution by amino-alcohol or incubation in Sudan Black B to remove lipofuscin as in Treweek et al., 2015 “Whole-body tissue stabilization and selective extractions via tissue-hydrogel hybrids for high-resolution intact circuit mapping and phenotyping” Nature protocols

-Part of the method section are highly detailed e.g. the anaesthetics used, while the more novel and important parts of the i.e. the staining is not described in detail to be able to replicate. E.g. which vein was injected, the tail vein and what volume and which concentrations are the antibodies in?

Could this be done with a lectin-Alexa647 or lectin-biotin and added strep-Alexa647 instead of targeting CD31, which may be less expensive?

- Since the Ethyl Cinnamate-based clearing procedure is new, I would recommend mentioning Ethyl Cinnamate in the abstract, so the reader can quickly identify the clearing method and differentiate it from other solvent-based methods, which would also promote the previous work of the authors.

- Abstract:

The following sentence is quite convoluted. Consider splitting it up:

“We discover and 3D-quantify distinguishable acute and late vascular I/R damage zones containing highly localized and spatially structured neutrophil infiltrates that are modulated upon cardiac healing. “

- The abbreviations CD31dim , CD31neg and CD31curly have not been explained nor defined in the text. Similar for Ly-6Gpos , TTCneg , F4/80pos

- “Interestingly, at d5 the majority of neutrophils was no longer associated with the border of the residual infarct-body but localized in the CD31curly volume (Supplementary Video 6). “ How many hearts were used to verify this and similar findings? It is unclear in the text.

- A minor note is that the combination of green and red images in figure 2b+c and 3b are not ideal for colour-blind readers. Replacing the red with e.g. magenta would solve this.

- Consider changing the title of the sub section in methods: “Light-sheet tissue processing and clearing“ into something more appropriate, like “BALANCE protocol for tissue clearing”

- Last sentence in this section: What is the concentration of ethyl cinnamate?

Contemporaneous 3D characterization of acute and chronic myocardial I/R injury and response

We thank the editors and the reviewers for their thorough consideration of our manuscript and their valuable input. We have addressed their specific remarks below on a point-by-point basis and revised the manuscript accordingly.

Editors' remark #1: In particular, please note that benchmarking to existing state-of-the-art clearing techniques and compelling evidence that your method is superior to these techniques will be a pre-requisite for further consideration of your paper.

Addressing this point, we added the new **Supplementary Table 3** (see also Reviewer #2, comment #6), putting our methods into perspective to multiple others applied in the field. Additionally, we i) conducted a direct comparison of CUBIC with BALANCE/LSFM that can be found in the new **Supplementary Fig. 12** (see also Reviewer #2, comment #3); ii) compared the BALANCE-mediated shrinkage factor to THF/DBE clearing (Reviewer #2, comment #7); iii) compared other bleaching approaches in the field with BALANCE in the new **Supplementary Fig. 2** (see also Reviewer #3, minor comment).

Furthermore, we demonstrate in the revised **Supplementary Fig. 3** that our method is applicable to other heavily autofluorescent murine organs and even human heart biopsies.

Editors' remark #2: Concerns raised regarding reproducibility should also be addressed.

We are confident that the reproducibility and applicability of BALANCE/LSFM are robust. To highlight this, we conducted benchmarking experiments with the gold-standard techniques in the field of ischemia/reperfusion (I/R) injury assessment. All of the following techniques, cardiac troponin I quantification, echocardiography, TTC staining and our BALANCE/LSFM approach were carried out within the same mouse, repeated using five mice. We were thus able to show highly significant correlation with BALANCE/LSFM and the other injury measurements, underlining the I/R injury representation of our method while simultaneously adding information on immune cell infiltrates (new **Supplementary Fig. 6** and Reviewer #1 concern #2). Hence, BALANCE/LSFM also allows extraction of more information per single animal, enhancing rigor and experiment robustness due to intra-animal correlations.

Furthermore, we enlarged the cohort of the 2D-vessel counting and the 3D filament models and substantially revised **Figure 2 and Supplementary Fig. 8** (also see Reviewer #3, comment #1.2).

We now precisely state the numbers of hearts used in each experiment and apologize for the confusion caused.

Reviewer #1:

The authors have developed a highly novel method to quantify myocardial cell injury and death in a 3-dimensional manner. The authors can also visualize the coronary circulation using the same technique. This technique is convincing and provides some new information. The studies have been performed in a very careful manner and there are no apparent flaws with this work.

We thank the reviewer for the kind comments regarding the overall concept of our studies.

The authors have compared this method with more traditional TTC and blue dye staining methods and have shown a good correlation. There are some advantages to this new method. However, the time and expense involved appears to be significant and it is not clear if this method can be widely adapted across many laboratories.

We thank the reviewer for this very important remark. Time and expense of experiments are indeed of high importance. In order to address these issues, we have now generated an additional table comparing the conventionally applied methods in the field of ischemia/reperfusion (I/R) injury in the murine heart¹ with our novel light sheet fluorescence microscopy (LSFM) approach (new **Supplementary Table 1** and as depicted below).

We here show that the hands-on time expense of our proposed LSFM procedure is comparable to the currently proposed standard methods in our hands.¹ However, none of the currently available approaches, including triphenyl tetrazolium chloride (TTC) staining, histology, immuno-histology and flow cytometry provide data that are comparable to LSFM in their type of site specific 3D information. At the same time, the combined experimental time needed to carry out all other singular methods in different animals can be significantly reduced by our multiplexing approach. Furthermore, we expect that in the near future, computer-based algorithms will further reduce analysis time as they will be able to automate much of the image recognition work that is key to extract meaningful biological data out of the LSFM image stacks.

BALANCE incorporates only few incubation steps, making sample handling comparably easy to perform. Technology, imaging devices, chemicals and expertise in our tissue clearing approach are commercially available, inexpensive and less toxic than other approaches in the field. Taken together, we believe that this technology and workflow would easily be available for use in many laboratories and in many different settings, species and tissues.

New **Supplementary Table 1** Comparison of assessed parameters, shortcomings, time-expense and difficulty of available techniques to evaluate myocardial ischemia/reperfusion (I/R) injury and associated parameters (2D – two-dimensional, AAR – area at risk, d – days, h – hours, LSFM – light sheet fluorescence microscopy, TTC – triphenyl tetrazolium chloride; * protocol time depends on automation processes)

Methods	Parameters assessed	Shortcomings	Total time until results	Hands-on time	Analysis time	Difficulty
TTC staining	Myocardial infarction parameters: Infarct size, AAR, remote zone (with blue dye), In pseudo-3D	No single cell resolution	2.5 h	1 h	0.25 h	Simple, parameters vary depending e.g. on operator and camera settings
Flow cytometry	Immune cell quantification and subset analysis	No precise assessment of I/R areas, no cell localization context	3.5 h	1.5 h	0.5 h	Simple to advanced analysis, depending on staining panel
(Immuno-) Histology	Infarct size, immune cell localization and quantification	Limited 3D spatial information due to thin 2D slices (<10µm); 3D information in sequential slices possible, but high time and cost expense and computational power involved	3-7 d	4 h	1-2 h, depending on level of analysis	Simple (H&E) to advanced (multicolor sequential immunohistochemistry)
BALANCE/ LSFM	Multiplexed infarction parameters (vascular damage, AAR and remote zone) with target cells in 3D; Identification of regions of interest for further processing (e.g. immunohistology)	Need for bleaching, due to whole organ imaging; target cell detection simple only with surface markers (i.v.-mediated staining, intracellular markers require more effort), limited multiplexing. 3D rendering software and high computational power required	1.5–4 d*	2.5 h	1.5 h for basic I/R injury parameters and cell segmentation; segment model or distance analysis more time consuming	Simple (infarction parameters and 3D immune cell localization) to advanced, depending on staining panel and post-processing

To further clarify these issues we added a statement in the revised version of the manuscript to the discussion section (page 8-9, lines 217-227):

*“We developed BALANCE for the fast 3D characterization of damage- and response-mechanisms in whole heart specimens. In contrast to iDISCO clearing, ethanol was used for sample dehydration in order to i) avoid high methanol toxicity and ii) reduce possible incompatibilities with following histological antibody stainings.² Also, long clearing times, e.g. in CUBIC³ or passive CLARITY⁴ were circumvented by the use of ECi (**Supplementary Table 2 and 3**). With the advent of robotics and artificial intelligence (AI) computation to automate clearing procedures and data analyses⁵, the usability and reliability of clearing methods and staining results will be further improved. Additionally, hands-on times are very similar to other conventionally applied approaches (**Supplementary Material and Supplementary Table 1 and 3** for a detailed comparison to conventional tools).”*

And to the supplementary information (page 17, lines 2-20):

“BALANCE in the field of cardiac ischemia (I/R) injury – multiplexing, rigor and expense

*In order to address the important issue of time and expense of experiments, we compared the conventionally applied methods in the field of I/R injury in the murine heart with our novel light sheet fluorescence microscopy (LSFM) approach (**Supplementary Table 1**). The hands-on time expense of our proposed LSFM procedure is comparable to each of the classical methods.¹ However, none of the currently available approaches, including triphenyl tetrazolium chloride (TTC) staining, histology, immuno-histology and flow cytometry provides LSFM-comparable data. LSFM combines I/R injury parameter assessment with cellular quantification and 3-D localization in the same heart. We therefore believe that with our proposed method, the robustness over experiments can be increased. At the same time, combined experimental time needed to carry out all other singular methods in different animals can be significantly reduced by our multiplexing approach. In the near future, computer-based algorithms will further reduce analysis time.*

BALANCE incorporates only few incubation steps, making sample handling easy to perform. Technology and chemicals for our tissue clearing approach are commercially available, inexpensive and less toxic than other approaches in the field. Taken together, we believe that this technology and workflow is available for use in many laboratories and in many different settings, species and tissues.”

The authors suggest that this method will help in the development of novel therapeutics to treat patients that suffer from an acute myocardial infarction and that infarct imaging has been a limitation in the translation of therapies to the clinic. This is simply not the case. The major limitations in the development of therapeutics to treat myocardial infarction in man are: (1) animal models (especially the mouse) do not closely mimic the human condition (2) lack of rigor in preclinical studies (3) myocardial reperfusion injury may not be a significant issue in man, (4) novel cardioprotective targets need to be identified to develop new therapeutics.

It is not clear that this new cardiac imaging modality addresses any of these limitations in the field. At present, the field is at a standstill and there are major questions regarding the utility of studying animal models of myocardial ischemia/reperfusion injury.

Cardioprotection summarizes all treatment options that reduce I/R injury.⁶ Although robust data are available for many novel therapeutic pathways in experimental studies, clinical data have been largely disappointing.⁷ However, risk of morbidity and mortality for patients with an acute myocardial infarction remains considerably high despite recent advances in percutaneous coronary interventions, platelet therapy and secondary prevention drugs.⁸ Therefore, there is a medical need for novel cardioprotective strategies. Given the body of negative clinical evidence as mentioned by the reviewer, the *European Union (EU)-CARDIOPROTECTION Cooperation in Science and Technology (COST) Action* has called for a re-evaluation of the aforementioned problems. Key results and future directions have recently been published^{1,7}. The underlying pathomechanisms in I/R injury are very complex and include e.g. reactive oxygen species, impaired nitric oxide signaling, cellular organelles (mitochondria), signaling pathways, interaction of heart cells (cardiomyocytes, endothelium, fibroblasts) with platelets and immune cells finally producing heart cell death via apoptosis and necrosis. Several promising strategies have been tested recently including beta-blocker metoprolol⁹⁻¹¹ and immune modulatory drugs^{12,13}. A common feature of these approaches is that they target multiple pathways involved in I/R injury rather than a single pathomechanism. Metoprolol targets both beta-receptors on cardiomyocytes to reduce myocardial energy consumption while inhibiting neutrophil-platelet interaction involved in microvascular obstruction⁹⁻¹¹. The authors of the recent position paper conclude that an ideal drug/approach for cardioprotection would target the cardiac vasculature, immune cells and cardiomyocytes⁷. With our proposed workflow, we provide a tool for the analysis of conventional I/R injury in conjunction with the characterization of the vascular injury several days after the initial I/R event. This can be related to immune responses in a spatial, volumetric resolution to study multiple targets in a bio-approach. We believe that BALANCE/LSFM can help to identify potential novel therapeutic options in the scope of the desired multi-target approach. To establish rigor in our model, we performed benchmarking as requested below. For this, our workflow has been tested against the established I/R analysis tools, which, in turn, have been performed in accordance with recently published guidelines for I/R injury^{1,14-17}. To address these issues we have now also added a corresponding statement to the discussion of our revised manuscript (pages 9 - 10, lines 237 – 250):

*“Arguably, many experimental cardioprotective approaches have yielded disappointing results when tested in clinical trials. One problem appears to be the lack of rigor in experimental studies¹. Hence we have conducted correlation analyses of our proposed analysis tool for the CD31^{neg} I/R injury against recently recommended standard approaches for experimental animal studies¹ and show a very good correlation (**Supplementary Fig. 6**). Furthermore, novel approaches are desired to improve the outcome of patients. It has been proposed that future therapeutic strategies should preferably target multiple pathways in I/R injury to battle the complex underlying signaling⁷. E.g. metoprolol was successfully tested in patients. It works by targeting beta-receptors on cardiomyocytes to reduce myocardial energy consumption while simultaneously inhibiting neutrophil-platelet interactions involved in microvascular obstruction⁹⁻¹¹. Thereby, LSFM studies could serve as a key tool to investigate the impact of such novel multi-target therapies on the whole heart.”*

This imaging modality is also limited since it requires sacrifice of the animal to perform the imaging. It appears that this method is limited to the murine heart.

We agree that, analogous to flow cytometry, triphenyl tetrazolium chloride (TTC) staining and immuno-histology, BALANCE/LSFM analysis is an end-point measurement. *In vivo* LSFM is only available in inherently clear organisms (e.g. zebra fish larvae).

However, the advantages of our method are its non-toxic nature and the combined assessment of injury and response mechanisms in the heart. Furthermore, our defined CD31^{curly} regions after 5 d of reperfusion match the I/R injury volumes after 24 h of reperfusion thus providing a potential proxy to the residuum of the vascular damage zone (**Fig. 2c**). Alternative techniques to evaluate specific immune cell infiltrates and I/R parameters in the whole heart *in vivo* are currently not available. In addition, our method correlates well with live heart function measurements as outlined in the specific comment to this reviewer's concerns #1-3.

Moreover, the method is not limited to the mouse heart. Instead, BALANCE can be used for reducing autofluorescence in any organ, enabling homogenous signal distribution. To demonstrate this further we have now included a new analysis of murine liver, which is particularly difficult to clear and image (new **Supplementary Fig. 3d**). Other investigated organs cleared with ECI include kidney⁵, gut¹⁸ and bone¹⁹.

To address the point of limitations across species, we now further set out to apply BALANCE to biopsies of the human heart (new **Supplementary Fig. 3e**). For this, we have processed a biopsy taken from the left atrial appendage (LAA) of an anonymized patient undergoing coronary bypass surgery. LAA tissue is routinely excised during bypass surgery to prevent cardio-embolic events. This analysis was approved by the local ethics committee of the Medical Faculty, University of Duisburg Essen, Germany (Ethics Approval Nr. 18-8527-BO). Subsequent processing and clearing were performed as described for murine hearts. This image demonstrates in a first feasibility approach that BALANCE has the potential to serve as a novel technique for the assessment of human tissue. A statement regarding this fact has been added to the manuscript text (page 5, lines 112 – 115):

*“Additionally, we could demonstrate the applicability of our novel clearing protocol to other murine tissues (e.g. liver, **Supplementary Fig. 3d**)^{5,18,19} and the human heart (left atrial appendage biopsy, **Supplementary Fig. 3e**).”*

We also added a paragraph to our methods section regarding the newly added human material (page 11, lines 278 – 284):

“Human studies

With written consent human left atrial appendage (LAA) biopsies were routinely excised during coronary bypass surgery presenting no further harm for the patient. The anonymized analysis was approved by the local ethics committee of the Medical Faculty, University of Duisburg-Essen, Germany (Ethics Approval Nr. 18-8527-BO) and the whole procedure was in accordance to the World Medical Association Declaration of Helsinki.”

Revised **Supplementary Fig. 3** (d) BALANCE applied to murine liver enhances sample transparency (*left*) and enables imaging of large tissue samples for quantitative purposes (*mid* and *right*). High contrast between vessel lumen and surrounding tissue allows direct rendering of the portal vein system from autofluorescence raw data (*mid*). (e) *Left*: BALANCE also enables clearing of a human left atrial appendage (LAA) biopsy and enhances sample transparency. *Mid*: The autofluorescence signal alone gives a structural overview and single cellular resolution (*right*). Scale bar values in μm . One square on the macroscopic images equals 2×2 mm. (ECi - ethyl cinnamate)

Concerns

(1) The authors have generated very small areas of infarction in the mouse heart. It is not clear why the infarct sizes are so small. Studies should be performed to create a variety of sizes of myocardial infarction and evaluate this new imaging method.

We thank the reviewer for this important remark and apologize for the misleading statements in our original version of the manuscript. We have also performed the additional experiments, which show a good correlation between standard I/R analysis techniques and our proposed work-flow from small to large areas of infarction.

In main **figure 1** we show an example of 10% infarction per area at risk (AAR). This was mentioned in the text, but mistakenly not referenced. We apologize for this error and corrected this also in the light of the additional experiments suggested by the reviewer (page 5, lines 134 - 135):

“Fig. 1b and c show a $CD31^{neg}$ to AAR injury of 10%, but the spectrum of small to larger infarction can be visualized as outlined below.”

In the manuscript we have furthermore demonstrated that our approach can delineate different areas of infarction in relation to AAR and remote zone as well as the TTC-staining method. In **Supplementary Fig. 7d and e** we show analogous to TTC staining that our proposed method is able to assess AAR and I/R injury along a range of different infarct sizes. However, TTC hearts and LSFM analyzed hearts were from different sets of experiments and not performed in one heart so that linearity cannot be analyzed and we propose that statistical test should be omitted.

New **Supplementary Figure 7 (d)** I/R injury size per AAR and AAR per heart as assessed by TTC and EB staining (n=5). **(e)** I/R injury size per AAR and AAR per heart as assessed by CD31 and FITC staining (n=5). Hearts depicted in (d) and (e) stem from separate cohorts.

In the original version of the manuscript we presented a good correlation between TTC^{neg} and CD31^{neg} I/R injuries in slices of the exact same heart analyzed in parallel by both methods (old **Supplementary Fig. 4e**). The overall size of the infarct per total heart was apparently comparably low (for a group of n=3 hearts). Of note, these have been calculated per heart and not per left ventricle of AAR. We next set out to generate a variety of different infarct sizes and compared TTC vs. LSFM in exactly the same slice and heart.^{16,17,20-22} In sum, along a range of small to large infarction, TTC and LSFM (TTC^{neg} and CD31^{neg}) areas show good correlation. These results have been integrated to our previous results now providing a significant linearity from small to large infarction concerning CD31^{neg} volumes and TTC^{neg} areas per slice and heart as requested (new **Supplementary Fig. 6**). Here we show a highly significant correlation between these two techniques, both per slice (r=0.9291, p<0.0001) and per heart (r=0.9858, p<0.0001), which was consistent for small and large infarct sizes. Thus for the range of infarction subsequent Bland-Altman analyses showed only small mean differences (-1.08% mean difference per slice and -0.75% mean difference per heart, new **Supplementary Fig. 6a**), indicating that TTC^{neg} and CD31^{neg} measure the same injury size. We also found inter-observer reliability for quantification of CD31^{neg} volumes to be low (<15% deviation from average, new **Supplementary Fig. 6b**).

We have added a statement to the manuscript relating to inter-observer variability and correlation based on the extended experiments (page 6, lines 150 – 162):

*“We correlated CD31^{neg} volumes to standard markers of myocardial damage, such as TTC staining, ejection fraction (EF) reduction and cardiac troponin I (cTNI) release, as recommended tools in experimental studies¹. We found a highly linear relationship between TTC^{neg} areas, CD31^{neg} volumes (both normalized to single slices as well as total hearts) and EF reduction (**Supplementary Fig. 6a and e**). Regions of abnormal left ventricular wall movement were located mostly in mid anterior, mid lateral and mid inferolateral zones, where*

TTC^{neg} and CD31^{neg} co-localized (**Supplementary Fig. 6c and d**). Plasma cTNI levels indicated establishment of infarction, but linearity was not statistically significant for the assessed infarction sizes (**Supplementary Fig. 6f**). Of note, infarct size determination by troponin assessment can be rather difficult in both, the clinical and experimental environment.^{1,23} Finally, a comparison of 3 operators regarding CD31^{neg} volume quantification yielded <15% variance of average (**Supplementary Fig. 6b**).”

f Measuring 4 I/R injury parameters in the same animal

Animal	EF reduction in %	TTC ^{neg} in %heart	CD31 ^{neg} in %heart	cTNI in ng/ml
#1	21.73	29.20	35.35	36.61
#2	15.70	19.49	21.58	58.19
#3	9.48	14.80	11.68	1.29
#4	5.86	12.70	14.40	36.49
#5	3.15	5.53	0.31	20.72
mean	30.66	16.34	16.66	11.19
s.d.	18.91	7.84	11.58	6.74

Correlation analyses of cTNI with other parameters assessed

	cTNI
EF reduction	r=0.45 p=0.45
TTC ^{neg}	r=0.40 p=0.51
CD31 ^{neg}	r=0.50 p=0.30

New **Supplementary Figure 6** Correlation of echocardiography, cardiac troponin I (cTNI) plasma levels, triphenyl tetrazolium chloride negative (TTC^{neg}) areas and CD31 negative (CD31^{neg}) volumes after 24 h of reperfusion in the same 5 mice. **(a) Top panel:** Correlation of CD31^{neg} volumes and TTC^{neg} areas per slice (*left*) and heart (*right*) of various infarct sizes (left: n=40 slices from 8 hearts; right: n=8 hearts; r- and p-values depicted). **Bottom panel:** Associated Bland-Altman analyses of CD31^{neg} volumes and TTC^{neg} areas per slice (*left*, -1.08% mean difference) and per heart (*right*, -0.75% mean difference) of correlations depicted above (dashed black lines mark 95% confidence intervals, solid black line marks zero bias, red dashed line marks mean difference). **(b)** Inter-operator CD31^{neg} volume quantification variances. (*Top*) Absolute quantified volume of 3 users in 5 randomly selected slices of different hearts subject to 45 min of ischemia and 24 h of reperfusion. (*Bottom*) The quantification results per user relative to the average (100%, dashed line) quantified volume (in comparison to *left*, slice #4 was excluded as no CD31^{neg} volume was measurable). **(c)** Original micrographs of echocardiographic imaging. Dashed lines indicate left ventricular lumen. White stars indicate zones of abnormal left ventricular wall movement. **(d)** Same I/R injury heart as in (c): exemplary TTC staining and subsequent light sheet fluorescence microscopy (LSFM)-based 3D reconstruction of the same slice. **(e)** Correlation of ejection fraction (EF) reduction with TTC^{neg} areas and CD31^{neg} volumes, respectively, per heart (n=5 hearts; r- and p-values depicted). **(f) Left:** raw data used for correlation of EF reduction, TTC^{neg} and CD31^{neg} with plasma cTNI levels (value for each animal given, n=5). **Right:** correlation of absolute cTNI levels with EF reduction, TTC^{neg} and CD31^{neg}. Scale bar values in μm .

(2) The authors should measure circulating and/or myocardial tissue levels of cardiac troponin I in these experiments.

(3) Similarly, the authors should also evaluate cardiac function and structure using echocardiography and correlate with infarct size.

The reviewer requests very important measurements, which were indeed omitted in our originally submitted manuscript.

To address these important points we have now included new echocardiographic measurements for 5 mice. The new data are shown in **Supplementary Fig. 6c - e**. Here, we observed a highly significant correlation between CD31^{neg} or TTC^{neg} sizes and ejection fraction (EF) reduction 24h after I/R injury, all of which was measured in the same hearts *in vivo* and after excision. We found that abnormalities in left ventricular wall movement were located to mostly mid anterior, mid lateral and mid inferolateral zones, which we determined to also be negative for TTC and CD31 staining (**Supplementary Fig. 6c and d**). The original manuscript text has been adapted accordingly (page 6, lines 150 – 156):

*“We correlated CD31^{neg} volumes to standard markers of myocardial damage, such as TTC staining, ejection fraction (EF) reduction and cardiac troponin I (cTNI) release as recommended tools in experimental studies¹. We found a highly linear relationship between TTC^{neg} areas, CD31^{neg} volumes (both normalized to single slices as well as total hearts) and EF reduction (**Supplementary Fig. 6a and e**). Regions of abnormal left ventricular wall movement were located mostly in mid anterior, mid lateral and mid inferolateral zones, where TTC^{neg} and CD31^{neg} co-localized (**Supplementary Fig. 6c and d**).”*

New **Supplementary Figure 6 (c)** Original micrographs of echocardiographic imaging. Dashed lines indicate left ventricular lumen. White stars indicate zones of abnormal left ventricular wall movement. **(d)** Same I/R injury heart as in (c): exemplary TTC staining and subsequent light sheet fluorescence microscopy (LSFM)-based 3D reconstruction of the same slice. **(e)** Correlation of ejection fraction (EF) reduction with TTC^{neg} areas and CD31^{neg} volumes respectively per heart (n=5 hearts; r- and p-values depicted).

Troponin is a very sensitive marker for the detection of myocardial injury and myocardial infarction.²³ To rule-out a myocardial infarction²⁴, clinical assessment of troponin is recommended for short periods. Myocardial infarction can be presumed with values above the 99th percentile. Indeed, our results indicate values above this threshold demonstrating that infarction was achieved by our mode. However, linearity between infarct sizes and troponin levels can be difficult²³ and linearity was not statistically significant in the additionally performed experiments (new **Supplementary Fig. 6f**). From practical experience in our lab, all 5 animals were well above the threshold of baseline mice (<0.02 ng/ml plasma cTNI, data not shown). These data are now discussed in the manuscript text as follows (page 6, lines 157 - 161):

*“Plasma cTNI levels indicated establishment of infarction, but linearity was not statistically significant for the assessed infarction sizes (**Supplementary Fig. 6f**). Of note, infarct size determination by troponin assessment can be rather difficult in both, the clinical and experimental environment^{1,23}.”*

New **Supplementary Figure 6 (f)** *Left*: raw data used for correlation of EF reduction, TTC^{neg} and CD31^{neg} with plasma cTNI levels (value for each animal given, n=5). *Right*: correlation of absolute cTNI levels with EF reduction, TTC^{neg} and CD31^{neg}.

References

- 1 Botker, H. E. *et al.* Practical guidelines for rigor and reproducibility in preclinical and clinical studies on cardioprotection. *Basic Res Cardiol* **113**, 39, doi:10.1007/s00395-018-0696-8 (2018).
- 2 Belle, M. *et al.* Tridimensional Visualization and Analysis of Early Human Development. *Cell* **169**, 161-173 e112, doi:10.1016/j.cell.2017.03.008 (2017).
- 3 Susaki, E. A. *et al.* Advanced CUBIC protocols for whole-brain and whole-body clearing and imaging. *Nat Protoc* **10**, 1709-1727, doi:10.1038/nprot.2015.085 (2015).
- 4 Tomer, R., Ye, L., Hsueh, B. & Deisseroth, K. Advanced CLARITY for rapid and high-resolution imaging of intact tissues. *Nat Protoc* **9**, 1682-1697, doi:10.1038/nprot.2014.123 (2014).
- 5 Klingberg, A. *et al.* Fully Automated Evaluation of Total Glomerular Number and Capillary Tuft Size in Nephritic Kidneys Using Lightsheet Microscopy. *J Am Soc Nephrol* **28**, 452-459, doi:10.1681/ASN.2016020232 (2017).
- 6 Yellon, D. M. & Hausenloy, D. J. Myocardial reperfusion injury. *N Engl J Med* **357**, 1121-1135, doi:10.1056/NEJMra071667 (2007).
- 7 Davidson, S. M. *et al.* Multitarget Strategies to Reduce Myocardial Ischemia/Reperfusion Injury: JACC Review Topic of the Week. *J Am Coll Cardiol* **73**, 89-99, doi:10.1016/j.jacc.2018.09.086 (2019).
- 8 Szummer, K. *et al.* Relations between implementation of new treatments and improved outcomes in patients with non-ST-elevation myocardial infarction during the last 20 years: experiences from SWEDEHEART registry 1995 to 2014. *Eur Heart J* **39**, 3766-3776, doi:10.1093/eurheartj/ehy554 (2018).
- 9 Ibanez, B. *et al.* Effect of early metoprolol on infarct size in ST-segment-elevation myocardial infarction patients undergoing primary percutaneous coronary intervention: the Effect of Metoprolol in Cardioprotection During an Acute Myocardial Infarction (METOCARD-CNIC) trial. *Circulation* **128**, 1495-1503, doi:10.1161/circulationaha.113.003653 (2013).
- 10 Garcia-Ruiz, J. M. *et al.* Impact of the Timing of Metoprolol Administration During STEMI on Infarct Size and Ventricular Function. *J Am Coll Cardiol* **67**, 2093-2104, doi:10.1016/j.jacc.2016.02.050 (2016).
- 11 Garcia-Prieto, J. *et al.* Neutrophil stunning by metoprolol reduces infarct size. *Nat Commun* **8**, 14780, doi:10.1038/ncomms14780 (2017).
- 12 Liberale, L. *et al.* Post-ischaemic administration of the murine Canakinumab-surrogate antibody improves outcome in experimental stroke. *Eur Heart J* **39**, 3511-3517, doi:10.1093/eurheartj/ehy286 (2018).
- 13 Gomez, D. *et al.* Interleukin-1beta has atheroprotective effects in advanced atherosclerotic lesions of mice. *Nat Med* **24**, 1418-1429, doi:10.1038/s41591-018-0124-5 (2018).
- 14 Hendgen-Cotta, U. B. *et al.* Nitrite reductase activity of myoglobin regulates respiration and cellular viability in myocardial ischemia-reperfusion injury. *Proc Natl Acad Sci U S A* **105**, 10256-10261, doi:10.1073/pnas.0801336105 (2008).
- 15 Luedike, P. *et al.* Cardioprotection through S-nitros(y)lation of macrophage migration inhibitory factor. *Circulation* **125**, 1880-1889, doi:10.1161/circulationaha.111.069104 (2012).
- 16 Rassaf, T. *et al.* Circulating nitrite contributes to cardioprotection by remote ischemic preconditioning. *Circ Res* **114**, 1601-1610, doi:10.1161/CIRCRESAHA.114.303822 (2014).
- 17 Totzeck, M., Hendgen-Cotta, U. B., French, B. A. & Rassaf, T. A practical approach to remote ischemic preconditioning and ischemic preconditioning against myocardial ischemia/reperfusion injury. *J Biol Methods* **3**, doi:10.14440/jbm.2016.149 (2016).
- 18 Zundler, S. *et al.* Three-Dimensional Cross-Sectional Light-Sheet Microscopy Imaging of the Inflamed Mouse Gut. *Gastroenterology* **153**, 898-900, doi:10.1053/j.gastro.2017.07.022 (2017).

- 19 Grüneboom, A. *et al.* A network of trans-cortical capillaries as mainstay for blood circulation in long bones. *Nature Metabolism*, doi:10.1038/s42255-018-0016-5 (2019).
- 20 Chen, J., Ceholski, D. K., Liang, L., Fish, K. & Hajjar, R. J. Variability in coronary artery anatomy affects consistency of cardiac damage after myocardial infarction in mice. *Am J Physiol Heart Circ Physiol* **313**, H275-H282, doi:10.1152/ajpheart.00127.2017 (2017).
- 21 Gao, X. M. *et al.* Microvascular leakage in acute myocardial infarction: characterization by histology, biochemistry, and magnetic resonance imaging. *Am J Physiol Heart Circ Physiol* **312**, H1068-H1075, doi:10.1152/ajpheart.00073.2017 (2017).
- 22 Yang, Z., Berr, S. S., Gilson, W. D., Toufektsian, M. C. & French, B. A. Simultaneous evaluation of infarct size and cardiac function in intact mice by contrast-enhanced cardiac magnetic resonance imaging reveals contractile dysfunction in noninfarcted regions early after myocardial infarction. *Circulation* **109**, 1161-1167, doi:10.1161/01.CIR.0000118495.88442.32 (2004).
- 23 Thygesen, K. *et al.* Fourth universal definition of myocardial infarction (2018). *Eur Heart J* **40**, 237-269, doi:10.1093/eurheartj/ehy462 (2019).
- 24 Zipes, D. P., Libby, P., Bonow, R. O., Mann, D. L. & Toaselli, G. F. *Braunwald's heart disease : a textbook of cardiovascular medicine*. 11th edn, (Elsevier, 2018).

Reviewer #2:

Merz et al. reported a pipeline for structural analyses of fluorescence-labeled myocardial I/R injury model in the murine hearts. The authors employed a commercial light sheet imaging system with the bleaching-augmented solvent-based non-toxic clearing (BALANCE) method to characterize and localize the infarcted myocardium in response to I/R injury. The authors demonstrate that BALANCE allows for detailed structure of myocardial infarction on the basis of ethyl-cinnamate as previously reported. While the authors compared BALANCE-LSFM to TTC-histology as a reference, the utility/advantage of their novel clearing method as compared to the current methodology was not articulated. While the authors provide the state-of-the-art imaging to characterize I/R injury patterns and to quantify acute and late vascular I/R damage zones, the imaging data are based on the commercialized LSFM, the clarity methodology was previously reported, and new biological findings are anticipated. Following are some comments to strengthen the manuscript.

We thank the reviewer for the appreciation of our work and for the opportunity to address the very important specific remarks as outlined below in order to strengthen the manuscript.

Comments

1. The authors used the commercial Ultramicroscope developed by LaVision BioTec to image most samples. The detailed procedures were not articulated in the Method section. Undefined are the total acquisition time, lateral resolution in the cross-sectional image, field of view, light sheet thickness, and image stitching. These factors are essential to assess the capability of this work. While the authors proposed to image the intact heart along the longitudinal axis, how to precisely position the long-axis along the scanning direction under the microscope was not articulated.

We apologize for not having given the exact details about our workflow. We have now added these aspects to our revised manuscript thereby adding detailed information on total acquisition time, lateral resolution, field of view (FOV), light sheet thickness and positioning of the hearts for quantitative imaging in the methods (pages 16 - 17, lines 424 - 443 of the revised version). We have used a FOV large enough to allow imaging of the specimens as a whole, so that image stitching could be omitted. We thus avoid significantly longer image acquisition times, signal inhomogeneity due to photobleaching and the use of stitching algorithms. We also included this information in the methods section of the revised manuscript (pages 16 - 17, lines 424 – 443 and below):

“For image acquisition, hearts were trapped, with apex and aorta horizontally aligned, in a commercially available sample holder (LaVision BioTec). To avoid damage or deformation of the sample, ECi-cleared 1% phytigel/ H₂O (CatNo: P8169-100G, Sigma Aldrich) blocks were used as buffers between tissue and plastic holder. Hearts were also rotated along the longitudinal axis, so that the knot, which remained in situ, faced downwards. This reduced blockage of excitation or emission light to a minimum. Whole-heart data sets were obtained with 2x total magnification (pixel size of 3.25 μm / pixel x,y, lateral resolution: 6.5 μm in x and y) with 10 μm z spacing between optical planes. Since the camera’s chip has 2560 x 2160 pixels, the field of view with this magnification amounts to 8.32 mm x 7.02 mm. It was thereby possible to image a whole mouse heart without multi-positioning, thus avoiding stitching algorithms while simultaneously reducing acquisition times. Sheet width was set to 4200 and

numeric aperture to 0.148, resulting in an approximate light sheet thickness of 4 μm in the horizontal focus. Illumination time was 350 ms, with enabled 8x dynamic focus in both left and right laser lines. Total acquisition time of one plane was thus, theoretically, 5.6 s. The vertical thickness of the hearts was generally smaller than 6 mm. This is the limiting factor due to the objective's working distance. Assuming a z stack as deep as 6 mm, this would have resulted in acquisition of 600 images x 4 channels = 2,400 images, which would theoretically have taken 3.73 h. However, in our practical experience one murine heart takes 4 - 6 h of acquisition time."

2. In reference to image processing, the authors used Imaris and ImageJ to reconstruct and to render the 3-D hearts. For vessel quantification, the authors applied previous reported method. Uncertain are the potential artifacts introduced by the rolling ball background subtraction and some other processing methods.

The preprocessing steps used were applied in order to reduce image noise, smooth out variations in the intensity of the labeling along vessels, as well as to significantly improve the contrast of the vessel regions against the background. These steps were found to be necessary since they significantly improved the tracing fidelity when applying the filament tracing algorithm in Imaris. In particular, when tracing in Imaris using the 'with loops' method, an initial simple thresholding was applied in order to segment vessels. Without smoothing, background subtraction and contrast enhancement, small vessels that are less intense were either inconsistently segmented due to variations in the background signal, were not distinctly segmented when close to larger, much brighter vessels, or were not segmented as an integral vessel, but were broken up due to inconsistencies in the labeling.

For the preprocessing in ImageJ, a standard Gaussian smoothing was first applied where the sigma parameter was chosen with a value of 2 μm , which is less than half the width of the smallest vessels. Next, a rolling ball background subtraction was applied. The rolling ball radius was set to a value that was approximately twice the radius of the largest vessels imaged, so as not to impact on the intensity distribution within vessels. This process in general can result in edge artifacts in the background signal where there are sharp changes in the intensity, in particular where background striping occurs in the light sheet images due to shadowing. However, because the resulting contrast for these background features was much lower than for the vessels, we could easily remove them during the thresholding process in Imaris. Finally, in order to further enhance the contrast of the vessels, whilst suppressing any non-vessel like features, we applied a multiscale Hessian-based vessel enhancement filter based on the method described by Frangi et al.¹ Therefore, we made use of a python script (vmtkimagevesselenhancement) obtained from the open source Vascular modelling ToolKit (VMTK) project (www.vmtk.org). The feature-based enhancement filter of Frangi is now a well-established tool for vessel segmentation.² In addition to the improved vessel contrast and non-vessel suppression, the Frangi filter also improved the continuity of the vessels by smoothing out inconsistencies in the vessel labeling along the length of the vessels. One known artifact of the filter is that it can lead to discontinuities at branching points, in particular for branching that occurs in a direction orthogonal to the main branch. Although we observed a reduction in the vessel intensity at these points, this effect occurred at a range very close to the branch point, and for our data it did not result in any discontinuities. Furthermore, we observed that the Frangi vessel filter can also affect the relative intensities of the vessel branches. Indeed, the intensities of smaller vessels were seen to be enhanced relative to larger vessels. In the Imaris filament tracer, when using a simple intensity threshold for segmentation, this could influence the evaluation of their relative diameters. However, we followed the initial construction of the filament model with a post-processing step to adjust the filament diameters

based on the relative contrast change across the diameter of the vessel. Here, a channel corresponding to the Gaussian smoothed and background subtracted original data was used. This additional post-processing step should at least correct for the general intensity mismatch. Although we have yet to fully evaluate the impact of both these effects on the evaluation of vessel diameter, we can conclude that they had little influence here, since - as stated in the manuscript - we only report on more robust parameters which depend on the vessel length or the number of branching points. Indeed, the effectiveness of this procedure for evaluating the vessel length has previously been demonstrated by us³, thereby showing good agreement with previously published results for the vessel length density in the brain cortex regions of mice. We have therefore found the Frangi vesselness filter to be an essential tool for the effective segmentation and modeling of microvessels, both in the brain and heart of mice, when used in combination with the Imaris filament tracer.

To specify these points, we have further added a section detailing the above-mentioned procedure to the revised manuscript methods section (pages 17 - 19, lines 463 - 509).

“Vessel quantification was performed as previously described³. Image data were first preprocessed, as a necessary step to reduce noise and improve vessel contrast, before applying a filament tracer model for vessel quantification in the 3D software package Imaris (BitPlane).

Briefly, image stacks were first preprocessed in the open source software ImageJ⁴ by applying a Gaussian smoothing, where a Gaussian sigma of 2 μm was chosen to be less than half the diameter of the smallest vessels. This was followed by a rolling ball background subtraction. In order not to affect the intensity distribution within the vessels themselves, a rolling ball radius of 20 μm was chosen, which was more than twice the diameter of the largest vessel. Any resulting edge artifacts from background variations could be removed during a later thresholding step in Imaris. The contrast of vessels was further enhanced using a python script (vmtkimagevesselenhancement) from the open source Vascular Modelling Toolkit project (VMTK: www.vmtk.org), which performed a multiscale Hessian-based Frangi vesselness filter.¹ The feature-based enhancement filter of Frangi is now a well-established tool for vessel segmentation.² In addition to the improved vessel contrast and non-vessel suppression, the Frangi filter also improved the continuity of the vessels by smoothing out inconsistencies in the vessel labeling along the length of the vessels. These preprocessing steps were found to be necessary for the fidelity of tracing smaller vessels when applying the filament-tracing algorithm in Imaris. In particular since, for the tracing of vessels, it was necessary to apply the ‘with loops’ filament tracing method, which uses an initial simple thresholding for segmenting the vessels. Without smoothing, background subtraction and contrast enhancement, small vessels that were less intense were either inconsistently segmented, due to variations in the background signal, not distinctly segmented when close to larger much brighter vessels, or were not segmented as an integral vessel, but were broken up due to inconsistencies in the labelling efficiency. It should also be noted that one known artifact of the Frangi filter is that it can lead to discontinuities at branching points. In particular, for branching that occurs in a direction orthogonal to the main branch. Although we observed a reduction in the vessel intensity at these points, this effect occurred at a range very close to the branch point, and for our data did not result in any discontinuities in the Imaris tracing. Furthermore, we also observed that the Frangi vessel filter can affect the relative intensities of the vessel branches. Indeed, the intensity of smaller vessels was seen to be enhanced relative to larger vessels. In the Imaris filament tracer, this could influence the evaluation of their relative diameters, due to the use of the simple intensity threshold method for segmentation. Indeed, in general, the application of a simple thresholding segmentation resulted in a relative mismatch in the filament diameters between small low intensity vessels and the larger brighter vessels, and also resulted in a mismatch for the filament path, calculated from the center lines of the

segmented vessels. Therefore, following the construction of the initial filament model, this mismatch was corrected in Imaris with two further processing steps. Firstly, a 'realignment' of the filament model was performed against the channel corresponding to the Gaussian smoothed and background subtracted vessel data. The same channel was then used to adjust the filament diameters, using an approach based on the relative contrast change across the diameter of the vessels. From the resulting filament model, parameters for the vascular length and branching points could be extracted."

3. The authors introduced a new chemical clearing approach or known as BALANCE for LSFM imaging. Unlike their previous publication in which ECi-based method was used (Reference 9), they incorporated a bleaching step to reduce the autofluorescence in this new protocol. However, this would probably quench the fluorescence in the transgenic mice, limiting the potential applications in other genetic models. For this reason, the precise quantification of fluorescence attenuation was not quantified.

We thank the reviewer for giving us the opportunity to address these very important issues. To address the role of bleaching for structural analysis of the heart, we have added a statement to the revised version of the manuscript (pages 4, lines 102 - 106):

"Applying BALANCE to our samples, tissue autofluorescence itself could be used to visualize the overall heart muscle structure allowing clear detection of cardiomyocytes and their nuclei (Supplementary Fig. 3a and b) at much higher optical resolution than previously possible^{5,6}."

The reviewer mentions the imaging of endogenous fluorophores. The heart naturally displays high autofluorescence, especially in short-wavelength channels e.g. 488 nm and 561 nm. Thus, the incorporation of a step reducing endogenous autofluorescence was required in order to allow homogenous, quantitative imaging. Of course this step is inclined to quench endogenously expressed fluorophores. However, without bleaching, it is impossible to homogeneously image tissue autofluorescence and particularly endogenous fluorophores in these wavelength domains (e.g. eGFP). In addition, in the absence of bleaching, hearts displayed not only a substantial illumination gradient, but also a significant decrease in signal to noise ratio in deeper regions of the tissue in all directions: x and y (mainly dependent on strength of excitation light and sample clarity) but also z (increasingly dependent on emission signal strength and sample clarity).

However, in order to test the reliability of our i.v.-mediated staining, we performed additional experiments. We assessed the 'Catchup mouse'⁷, in which neutrophils express tdTomato. We compared a non-bleached Catchup heart cleared with ECi with hearts cleared using BALANCE or CUBIC⁸, an aqueous buffer clearing technique known to preserve endogenous fluorescence well (new **Supplementary Fig. 12**). In these new experiments, CUBIC showed an inferior macroscopic sample clarity compared to BALANCE. Moreover, in our hands Ly-6G and CD31 i.v.-mediated labeling were nearly undetectable when using CUBIC, while endogenous fluorophores were nicely visible. On the other hand, BALANCE, as expected, quenched endogenous fluorescence along with tissue autofluorescence, but enabled homogenous imaging of the Ly-6G i.v. labeling. Remarkably, using the ECi protocol without bleaching⁹, we detected both, Ly-6G staining and the tdTomato signal. However, the latter displayed a much weaker signal-to-noise ratio. The total abundance of Ly-6G signals in these samples was much higher than visible tdTomato signals, further demonstrating the signal-to-noise problem that the highly autofluorescent muscle tissue poses.

Taken together, BALANCE should clearly be favored for exogenous labels together with the desired analysis of tissue structure from autofluorescence as required for this I/R injury analysis workflow. We have also added a statement detailing this to the revised manuscript (page 5, lines 115-119):

“Note that BALANCE was designed to maximize the signal quality of intravenous (i.v.)-mediated staining using synthetic fluorophores, allowing cell identification with high precision against a highly autofluorescent background. Homogenizing tissue autofluorescence with peroxide may quench endogenously expressed fluorophores as well (Supplementary Fig. 12).”

New Supplementary Figure 12 Endogenous fluorescence and i.v.-mediated staining in organic- and water-based clearing. Comparison of (a) CUBIC, (b) BALANCE and (c) ECI-only clearing in their capabilities of preserving endogenous (tdTomato, green) and artificial fluorophore (anti-mLy-6G AF647, magenta) fluorescence in Catchup mouse⁷ hearts (neutrophils tdTomato positive) after myocardial ischemia/reperfusion (I/R) injury. (d) Magnification of ROIs depicted in (c, white rectangles). Scale bar values in μm . One square on the macroscopic images (left) equals 2x2 mm.

Finally, deep imaging (deeper than 2-3 mm) in highly autofluorescent organs using short wavelengths is associated with increasing blurriness, even though the tissue is bleached and cleared.¹⁰ Therefore, we only used FITC (fluorescein isothiocyanate) albumin filling in the blue channel (488 nm excitation, 525/50 nm emission), which yields an extremely bright signal for area at risk (AAR) rendering. The autofluorescence signal attenuation was quantified in **Supplementary Fig. 1c** and additional measurements are now provided in the new **Supplementary Fig. 2**.

Supplementary Figure 1 (c) Signal distribution of ethyl cinnamate (ECi)-cleared, unbleached or bleached hearts in LSFM in different channels, single wavelength excitation and filtered detection range given in wavelength/range, detecting autofluorescence. The region of interest (ROI) in the left ventricular wall (LVW) shows high autofluorescence at the edge of unbleached hearts. This peak is lowered, together with an overall lower autofluorescence intensity in bleached hearts. The homogenization of autofluorescence is most prominent in the short wavelength channels, unlocking those for quantitative imaging and general thresholding. Scale bar values in μm .

4. In reference to Comment #3, the authors used IV injections of fluorescently labeled antibodies. Currently available techniques allow for clearing of heart, followed by labeled-antibody exposure and signal detection. To demonstrate the utility of their technique, the authors may consider the transgenic lines, such as *cm1c-gfp*, to demonstrate the absence of photobleaching and attenuation of the fluorescent signal.

Other protocols label cells after sample fixation via whole mount staining (iDISCO).¹¹ However, we chose to implement i.v. mediated labeling to significantly speed up the protocol and ensure labeling of cells in their original state.

Furthermore, as proposed, we have now conducted additional experiments in fluorescent protein-transgenic mice using additional markers and incorporated the results in the revised version of our manuscript (compare response to point #3 above). Further data are presented in new **Supplementary Fig. 12c** and **d**. Here, we found that all tdTomato+ cells in the tissue were also positive for Ly-6G labeling, strengthening the applicability of this antibody delivery route for immune cells and surface markers. However, the signal-to-noise ratio of tdTomato was low, with signal intensities matching muscle autofluorescence. Using BALANCE, we lower the tissue autofluorescence in order to enhance sample clarity and signal homogeneity of synthetic dyes but quench endogenous fluorescence. This is now summarized in the new **Supplementary Fig. 12** and stated in the revised manuscript (page 5, lines 115 – 119):

“Note that BALANCE was designed to maximize the signal quality of intravenous (i.v.)-mediated staining using synthetic fluorophores, allowing cell identification with high precision against a highly autofluorescent background. Homogenizing tissue autofluorescence with peroxide may quench endogenously expressed fluorophores as well (Supplementary Fig. 12).”

Statement in the supplementary information (page 18, lines 33 – 47):

“Endogenous fluorophores

*We used the Catchup mouse model⁷ expressing tdTomato in neutrophils to benchmark our i.v.-mediated Ly-6G staining in terms of cellular labeling efficiency. Clearing an infarcted murine Catchup heart with ethyl cinnamate (without peroxide treatment) allowed for visualization of both, i.v.-mediated Ly-6G labeling and tdTomato signal (**Supplementary Fig. 12c and d**)⁹. Here, we found that all tdTomato positive cells in the tissue were also positive for Ly-6G labeling, strengthening the applicability of this antibody delivery route for immune cells and surface markers. However, the signal-to-noise ratio of tdTomato was low, with signal intensities matching muscle autofluorescence. Using BALANCE, we lowered the tissue autofluorescence in order to enhance sample clarity and signal homogeneity of synthetic dyes. However, hereby endogenously expressed tdTomato was quenched as well (**Supplementary Fig. 12b**). Additionally, when using the CUBIC protocol⁶, we were able to visualize the endogenous fluorescence, but observed less sample clarity and lost the artificial Ly-6G staining completely (**Supplementary Fig. 12a**; this caveat is described in the original paper and can be potentially circumvented by further establishment).”*

Therefore, we have not conducted any experiments with GFP-expressing mice.). Furthermore, autofluorescence in the 488nm channel is even higher compared to 561nm excitation.

5. Please demonstrate applicability of current method using multi-channel excitation in transgenic mice labeled with 2 or more fluorophores.

As detailed above (comments #3 and 4) it is not possible to generate meaningful imaging data of endogenously fluorescent cells/structures using BALANCE/LSFM. However, BALANCE is definitely able to achieve multiplexed imaging of up to 3 exogenously added fluorescent signals and autofluorescence simultaneously (e.g. FITC, autofluorescence, AlexaFluor 647 and 790; see **Revision Figure 1** and **Supplementary Movie 1, 2, 3 and 7**). Using other optical setups (lasers, filters, detectors) and different synthetic colors it is not principally impossible to also image even more distinct signals simultaneously with BALANCE/LSFM. Hence, we believe that color labelling by injection of fluorescent tracers is much more flexible than the use of multi-locus transgenic mice.

Revision Figure 1 Four color imaging after the BALANCE protocol. Visualization of FITC-albumin filling (488 nm laser line), tissue autofluorescence (561 nm laser line), Ly-6G-AF647 (639 nm laser line) and CD31-AF790 (785 nm laser line) for analysis of myocardial infarction in a murine heart.

6. The advantages of proposed BALANCE method would be strengthened by comparing with other tissue clearing methods, including CLARITY, iDISCO, CUBIC and SWITCH. Please clarify the reason for choosing this specific method in the Discussion section.

We thank the reviewer for this suggestion. The methods mentioned by the reviewer have recently been compared in depth in two comprehensive studies^{10,12}. We have now added the same features with regard to BALANCE to these methods and summarized these data in the new **Supplementary Table 3**. Collectively, this analysis shows that most protocols, including CUBIC and CLARITY, are not as efficient as our approach to clear the heart. This particularly refers to the required lowering of autofluorescence and to homogenizing the signal, especially in lower wavelength channels. Other downsides of the evaluated protocols were toxicity, complexity of fluid handling and procedure, special equipment required, length of the protocol and handling of the sample after clearing. With BALANCE, we have established a protocol that combines non-toxic reagents, short handling and protocol times, less complexity and superior signal homogenization than the other clearing protocols currently available. We addressed these important points in the discussion section now (see pages 8-9, lines 218 – 227):

*“In contrast to iDISCO clearing, ethanol was used for sample dehydration in order to i) avoid high methanol toxicity and ii) reduce possible incompatibilities with following histological antibody stainings.¹³ Also, long clearing times, e.g. in CUBIC⁸ or passive CLARITY¹⁴ were circumvented by the use of ECI (**Supplementary Table 2 and 3**). With the advent of robotics for sample handling and artificial intelligence (AI) computation for data analyses⁹, the usability and reliability of clearing methods and staining results will be further enhanced. Additionally,*

hands-on times are very similar to other conventionally applied approaches (see **Supplementary Table 1 and 3** for a detailed comparison to conventional tools).”

A comparison with CUBIC clearing can be found in the new **Supplementary Figure 12**, as response to comment #3.

New **Supplementary Table 3**: Comparison of established clearing methods with the BALANCE protocol. Evaluation of CUBIC¹⁰, CLARITY¹⁰, iDISCO¹² and SWITCH¹² as previously published.

	Macroscopic sample clarity	Autofluorescence homogenization	Endogenous fluorophores preserved	Tissue alteration	Sample integrity	Incubation time	Characteristics
ECi	Moderate - brown	No	Yes	Slight shrinkage (~20%)	Stiff	1 d	Non-toxic
BALANCE	Clear - light yellow	Yes	No	Slight shrinkage (18% ± 2)	Stiff	1.5 d	Non-toxic
CUBIC¹⁰	Moderate – light brown	Not tested	Yes	Expansion	Spongy	14 d	Non-toxic
CLARITY¹⁰	Moderate – light brown	Not tested	Yes	Expansion	Spongy	10 d	Toxic
iDISCO¹²	Clear – light brown	Yes	Yes	Shrinkage	Stiff	Hours – days	Toxic
SWITCH¹²	Clear – light yellow	Yes	No	Slight expansion	Not tested	Days	Toxic

7. While the solvent-based clarity method is well-known to cause tissue shrinkage, whether the tissue shrinkage factor based on BALANCE is superior to other solvent-based method such as iDISCO (Cell, 2017) remains unclear.

We determined the shrinkage from naïve tissue to ECi cleared heart muscle tissue using our BALANCE protocol to be 18% ± 2 (mean ± s.d.) for heart slices in every dimension, as now correctly stated in the methods section (page 19, lines 514 – 516). The low tissue shrinkage in the heart is most likely based on the very dense muscle tissue. However, using 3DISCO also Belle et al¹³ report a 20-40% shrinkage of human embryos that is homogenous throughout the sample. We now performed additional experiments to determine the tissue shrinkage factor for the heart using THF/DBE clearing. We measured 55% ± 3 (mean ± s.d.) shrinkage from native to cleared tissue, similar to values found by Orlich and Kiefer.¹⁰ From these analyses it appears, that BALANCE induces less tissue shrinkage than iDISCO.

We have added a statement to the manuscript text on page 19, lines 514 - 516:

“Tissue shrinkage factor was 0.82 ± 0.02 (or 18% ± 2; mean ± s.d.) and every spatial dimension was corrected using this factor.”

8. Please provide a detailed protocol for BALANCE for the light sheet imaging community.

We now included more detailed information in the methods sections “BALANCE protocol for tissue clearing” and “Light sheet fluorescence microscopy and image processing”. The details can be found methods section of the revised version of the manuscript (pages 13 - 14, lines 345 - 362 and below):

“BALANCE protocol for tissue clearing

Perfused hearts were immersed in 4% paraformaldehyde (w/v, PFA, CatNo. 10195, Morphisto) in PBS for chemical fixation while standing for 4 h at 4 °C in 15 ml tubes. For all following steps, hearts were kept at 4 °C, always agitated, transferred only to pre-cooled solutions in 15 ml tubes and kept protected from light. Following fixation, hearts were dehydrated in an ascending ethanol (CatNo. T171.3; Roth) series in ddH₂O (v/v) of 50%, 70% and 100% (CatNo. 9065.2, Roth) for at least 4 h while shaking. Samples can also be stored longer at each step without loss of staining quality.

Subsequently, samples were bleached for 4 h in freshly prepared 5% (v/v) hydrogen peroxide (CatNo. 349887; Sigma Aldrich) and 5% (v/v) dimethyl sulfoxide (CatNo. 4720.3, Roth) in 100% ethanol. This step can potentially induce generation of pressure within the tube. Bleaching was carried out to enhance sample clarity, enabling homogenous imaging in lower wavelength channels while preserving artificial fluorophore fluorescence (**Supplementary Fig. 1**). After a washing step of at least 4 h in 100% ethanol, samples were warmed to room temperature (RT) for 5 min before transfer into 7 ml pure (99%) ethyl cinnamate (ECi, CatNo. 112372, Sigma Aldrich) in a glass vial for at least 4 h prior to imaging. Samples were kept at in the dark until and after imaging.

Also, see page 16 – 17, lines 415 – 458:

Light sheet fluorescence microscopy and image processing

Samples were imaged using an Ultramicroscope II and ImSpector software (both LaVision BioTec). The microscope is based on a MVX10 zoom body (Olympus) with a 2x objective and equipped with a Neo sCMOS camera (Andor). For image acquisition, cleared samples were immersed in ECi in a quartz cuvette and excited with light sheets of different wavelengths (488 nm, 561 nm, 639 nm and 785 nm). The following band-pass emission filters (mean nm / spread) were used, dependent on the excited fluorophores: 525/50 for FITC; 595/40 for AF594 or autofluorescence; 680/30 for AF647 and 835/70 for AF790. For quantitative imaging, whole hearts were imaged along the longitudinal axis. For image acquisition, hearts were trapped, with apex and aorta horizontally aligned, in a commercially available sample holder (LaVision BioTec). To avoid damage or deformation of the sample, ECi cleared 1% phytigel/H₂O (CatNo: P8169-100G, Sigma Aldrich) blocks were used as buffers between tissue and plastic holder. Hearts were also rotated along the longitudinal axis, so that the knot, which remained in situ, faced downwards, thus reducing blockage of excitation or emission light to a minimum. Whole-heart data sets were obtained with 2x total magnification (pixel size of 3.25 μm / pixel x,y, lateral resolution: 6.5 μm in x and y) with 10 μm z spacing between optical plains. Since the camera's chip has 2560 x 2160 pixels, the field of view with this magnification amounts to 8.32 mm x 7.02 mm. Thus, it was possible to image a whole mouse heart without having to use multi-positions, avoiding stitching algorithms and reducing acquisition times. Sheet width was set to 4200 and numeric aperture to 0.148, resulting in an approximate light sheet thickness of 4 μm in the horizontal focus. Illumination time was 350 ms, with enabled 8x dynamic focus in both left and right laser lines. Total acquisition time of one plane was thus, theoretically 5.6 s. Vertical thickness of the hearts was generally smaller than 6 mm. This is the limiting factor due to the objective's working distance. Assuming a z stack as deep as 6 mm, this would have resulted in acquisition of 600 images x 4 channels = 2400 images, which would theoretically have taken 3.73 h. However, our practical experience showed 4 - 6 h of acquisition time of one murine heart due to stage movements. For regions of interest (ROIs), magnifications are indicated in the figure legends.

See <https://www.lavisionbiotec.com/products/UltraMicroscope/specification.html> for zoom factors, corresponding numerical apertures and resolutions. To avoid photobleaching during imaging, the longest wavelength was imaged first.

16bit OME.TIF stacks were converted (ImarisFileConverterx64, Version 9.2.0, BitPlane) into Imaris files (.ims). 3D reconstruction and subsequent analysis was done using Imaris software (BitPlane). Heart surface (25 μm grain size) and volume was determined using

surface creation algorithms. All surface tracings (AAR, CD31^{neg}, CD31^{curly}, vessels) were based on the contour tracing tool and carried out manually/semi-automatically. For surface creation, each (vessels), every 5th (CD31 tracings) or 10th (AAR) image was traced. Surfaces were created with maximum resolution (2160 x 2560 pixel) and preserved features, to ensure matching of the traced lines with the surface border. Discrimination of arteries and veins was done by their respective location within the heart muscle and the signal intensity of the CD31 staining (CD31^{high} for arteries, CD31^{low} for veins).”

References

- 1 Frangi, A. F., Niessen, W. J., Vincken, K. L. & Viergever, M. A. Multiscale vessel enhancement filtering. *Medical Image Computing and Computer-Assisted Intervention - Miccai'98* **1496**, 130-137 (1998).
- 2 Lesage, D., Angelini, E. D., Bloch, I. & Funka-Lea, G. A review of 3D vessel lumen segmentation techniques: models, features and extraction schemes. *Med Image Anal* **13**, 819-845, doi:10.1016/j.media.2009.07.011 (2009).
- 3 Lugo-Hernandez, E. *et al.* 3D visualization and quantification of microvessels in the whole ischemic mouse brain using solvent-based clearing and light sheet microscopy. *J Cereb Blood Flow Metab* **37**, 3355-3367, doi:10.1177/0271678x17698970 (2017).
- 4 Schneider, C. A., Rasband, W. S. & Eliceiri, K. W. NIH Image to ImageJ: 25 years of image analysis. *Nat Methods* **9**, 671-675 (2012).
- 5 Lee, S. E. *et al.* Three-dimensional Cardiomyocytes Structure Revealed By Diffusion Tensor Imaging and Its Validation Using a Tissue-Clearing Technique. *Sci Rep* **8**, 6640, doi:10.1038/s41598-018-24622-6 (2018).
- 6 Fei, P. *et al.* Cardiac Light-Sheet Fluorescent Microscopy for Multi-Scale and Rapid Imaging of Architecture and Function. *Sci Rep* **6**, 22489, doi:10.1038/srep22489 (2016).
- 7 Hasenberg, A. *et al.* Catchup: a mouse model for imaging-based tracking and modulation of neutrophil granulocytes. *Nat Methods* **12**, 445-452, doi:10.1038/nmeth.3322 (2015).
- 8 Susaki, E. A. *et al.* Advanced CUBIC protocols for whole-brain and whole-body clearing and imaging. *Nat Protoc* **10**, 1709-1727, doi:10.1038/nprot.2015.085 (2015).
- 9 Klingberg, A. *et al.* Fully Automated Evaluation of Total Glomerular Number and Capillary Tuft Size in Nephritic Kidneys Using Lightsheet Microscopy. *J Am Soc Nephrol* **28**, 452-459, doi:10.1681/ASN.2016020232 (2017).
- 10 Orlich, M. & Kiefer, F. A qualitative comparison of ten tissue clearing techniques. *Histol Histopathol* **33**, 181-199, doi:10.14670/HH-11-903 (2018).
- 11 Renier, N. *et al.* iDISCO: a simple, rapid method to immunolabel large tissue samples for volume imaging. *Cell* **159**, 896-910, doi:10.1016/j.cell.2014.10.010 (2014).
- 12 Yu, T., Qi, Y., Gong, H., Luo, Q. & Zhu, D. Optical clearing for multiscale biological tissues. *J Biophotonics* **11**, doi:10.1002/jbio.201700187 (2018).
- 13 Belle, M. *et al.* Tridimensional Visualization and Analysis of Early Human Development. *Cell* **169**, 161-173 e112, doi:10.1016/j.cell.2017.03.008 (2017).
- 14 Tomer, R., Ye, L., Hsueh, B. & Deisseroth, K. Advanced CLARITY for rapid and high-resolution imaging of intact tissues. *Nat Protoc* **9**, 1682-1697, doi:10.1038/nprot.2014.123 (2014).

Reviewer #3:

In this manuscript, Merz and colleagues characterise the 3-dimensional extend of acute and chronic myocardial injury and response. They provide new data on cardiac ischemia and propose a promising new method for further work in that field and tissue clearing. The paper is well written and illustrated with figures of the methodology and results. This work will likely be helpful for future researchers in cardiovascular disease and stroke. The movies are also beautiful and instructive. I have some major and minor concerns about the study:

We thank the reviewer for his/her kind consideration of our work and methodology.

Major:

1) The primary weaknesses of this study is the 1) lack of hypothesis, and 2) potentially anecdotal nature. I could not find anything in the text about sample size. How many animals were used? On page 12 it said: “8 slices of the right ventricular wall (RVW), the intraventricular septum (IVS) and the left ventricular wall (LVW) of the same heart (n=1) were analyzed. “ In figure 2 caption: “volume normalized to heart muscle volume (including appendages and right ventricle (RV)) over time (n=5; median; two-sided Mann-Whitney test, p-value depicted).” What is n here? Number of animals?

We thank the reviewer for his/her comments and apologize for not making these aspects clearer in the originally submitted version of our manuscript.

Comment 1.1

We hypothesize that our workflow can characterize myocardial ischemia/reperfusion (I/R) injury in conjunction with response mechanisms, particularly immune signaling. The precise goals were: (i) development of a readily applicable, non-toxic workflow with chemicals and tools that are commercially available, (ii) benchmarking against standard I/R histology techniques in compliance with recent guideline recommendation¹, (iii) volumetric quantification of I/R bodies, (iv) long-term assessment of I/R injury following days after reperfusion (assessment of CD31^{curly} at 5d) and (v) relation of I/R injury zones to immune response mechanisms. We now added a paragraph to the Introduction section of the revised version of the manuscript detailing these points (page 3, lines 76 – 79 and page 4, lines 87 - 92 and below).

“Therefore, we hypothesize that LSFM is capable of characterizing myocardial ischemia/reperfusion (I/R) injury in conjunction with significant I/R injury response mechanisms, particularly immune cell infiltration. [...] In summary, we aimed at (i) developing a readily applicable, non-toxic workflow with chemicals and tools that are commercially available, (ii) benchmarking this work flow against standard I/R histology techniques in compliance with recent guideline recommendations¹, (iii) quantifying I/R injury parameters in 3D, (iv) assessing the long-term impact of I/R injury following days after reperfusion and (v) relating the I/R injury zones to immune response mechanisms.”

Comment 1.2

The manuscript has been adapted at the requested part and throughout the entire revised version. In the original manuscript, we only analyzed 8 slices of the RVW, IVS and LVW in 1 heart (hence n=1) for representative demonstration. We have now expanded this experiment (new **Supplementary Fig. 8a**) by 2 more hearts (overall n=3 hearts) to make our finding more robust. We observe a wide range of 2D counting differences within one heart. Reasons for this are i) the differential directionality of vessels (point vs filament count) within one field of view and ii) the vessels that surface multiple times in one image (new **Supplementary Figure 8a-**

c). Based on this more comprehensive analysis we find, that, as opposed to the analysis in just one heart, there are no significant differences between the compartments, when analyzing several hearts (LVW, IVS, RVW). Furthermore, we have added FRANGI/Filament model quantifications (overall n=3 hearts) (new **Fig. 2e**) and show a statistical analysis. Overall, there appears to be a clear, yet insignificant trend that the vessel length density within the LVW of basal hearts is higher compared to the CD31^{curly} region after 5d of reperfusion (n=3; p=0.1, two-sided Mann-Whitney test) principally confirming our original observation. N is always the number of animals / hearts as now articulated precisely throughout the manuscript.

New **Supplementary Figure 8** 2D and 3D cardiac vessel measurements. **(a)** Quantification of CD31 positive (CD31^{pos}) vessels in optical slices at different z positions of the right ventricular wall (RVW), intraventricular septum (IVS) and left ventricular wall (LVW) of the murine heart (n=3 mice, each dot represents one counted ROI; the respective median, as well as a median summary for each compartment is shown). **(b)** Exemplary field of view showing diverse vessel directionality in LVW. **(c)** Images show that counting in 2D (e.g. optical slice) is difficult and results in repeated measures of the same vessel (yellow arrows) visualized by a maximum intensity projection (MIP) of 25 μm z depth (magenta). Magnification: 6.4x.

New **Figure 2 (e)** Vessel length density (VLD, *left*) and interbranch distance (IBD, *right*) in baseline and CD31^{curly} regions (ROIs from 3 hearts; median). Scale bar values in μm.

These new observations have been added to the revised manuscript text (page 7, lines 168 – 179):

*“2D quantification of vessel densities revealed no significantly different values within the heart muscle, but high variances concerning the field of view in the same heart region (**Supplementary Fig. 8a**). We observed a variability in directionality of vessels, which seemed to be linked to localization within the muscle (**Supplementary Fig. 8b**). In addition, 3D analysis of the left ventricular wall displayed continuous tracts of capillaries surfacing multiple times in one image, which might explain counting errors in 2D immuno-histology (**Supplementary Fig. 8c**). We next employed a filament tracing pipeline² to quantify length and complexity of the*

vasculature in 3D. In the left ventricular wall we found a vessel length density (VLD) of 3,145 mm per mm³ and an interbranch distance (IBD) of 64 μm (n=3; median) reflecting a >2x higher VLD compared to brain (**Fig. 2d and e and Supplementary Fig. 8e**)².”

And page 7, lines 189 – 192:

“Compared to healthy myocardium, filament tracing in curly areas revealed a trend for lower VLD (1,890 mm/mm³; n=3, median) and smaller IBD (50 μm; n=3, median) values, which principally confirms the initial observation of a shorter and more complex vascular network at sites of injury (**Fig. 2d and e**).”

2) Since there is no hypothesis, the objectives should be clearly stated. It seems that part of the objectives of the study primarily to improve the clearing of cardiac tissue, that has “extremely high autofluorescence” in the previously applied method. Nevertheless, if this is the primary goal, why not perform a systematic analysis of the existing clearing methods, rather than ad hoc inventing a new method? Or at least mention other clearing methods and their shortcomings in this context?

We have now precisely laid out our goals as described above (Q1). As requested we have performed a substantial benchmarking of BALANCE against existing methods and report on the individual features we could identify (see also new **Supplementary Fig. 2 and 12**).

We established BALANCE to overcome the current problems of heart clearing, as previously summarized.^{3,4} We adapted their evaluation approach for the ECi and BALANCE protocols in **Supplementary Table 3**. Most protocols, including CUBIC and CLARITY, are not as efficient as our approach to clear the heart, lowering autofluorescence and homogenizing the signal, especially in lower wavelength channels. Other downsides of the evaluated protocols were toxicity, complexity of fluid handling and procedure, special equipment required, length of the protocol and handling of the sample after clearing. With BALANCE, we have established a protocol that combines non-toxic reagents, short handling and protocol times, less complexity and superior signal homogenization than the other clearing protocols currently available. We addressed these important points in the discussion section now (see pages 8-9, lines 218 – 227):

“In contrast to iDISCO clearing, ethanol was used for sample dehydration in order to i) avoid high methanol toxicity and ii) reduce possible incompatibilities with following histological antibody stainings.⁵ Also, long clearing times, e.g. in CUBIC⁶ or passive CLARITY⁷ were circumvented by the use of ECi (**Supplementary Table 2 and 3**). With the advent of robotics and artificial intelligence (AI) computation automating clearing procedures and data analyses⁸, the usability and reliability of clearing methods and staining results will be further enhanced. Additionally, hands-on times are very similar to other conventionally applied approaches (see **Supplementary Table 1 and 3** for a detailed comparison to conventional tools).”

Supplementary Table 3: Comparison of established clearing methods with the BALANCE protocol. Evaluation of CUBIC³, CLARITY³, iDISCO⁴ and SWITCH⁴ as previously published.

	Macroscopic sample clarity	Autofluorescence homogenization	Endogenous fluorophores preserved	Tissue alteration	Sample integrity	Incubation time	Characteristics
ECi	Moderate - brown	No	Yes	Slight shrinkage (~20%)	Stiff	1 d	Non-toxic
BALANCE	Clear - light yellow	Yes	No	Slight shrinkage (18% ± 2)	Stiff	1.5 d	Non-toxic
CUBIC ³	Moderate – light brown	Not tested	Yes	Expansion	Spongy	14 d	Non-toxic
CLARITY ³	Moderate – light brown	Not tested	Yes	Expansion	Spongy	10 d	Toxic
iDISCO ⁴	Clear – light brown	Yes	Yes	Shrinkage	Stiff	Hours – days	Toxic
SWITCH ⁴	Clear – light yellow	Yes	No	Slight expansion	Not tested	Days	Toxic

Minor:

-Regarding the bleaching with peroxide, it is an effective approach, but also very harsh for the tissue and may damage antigens. I am missing the data or at least the argumentation for selecting peroxide-bleaching compared to other methods such as heme elution by amino-alcohol or incubation in Sudan Black B to remove lipofuscin as in Treweek et al., 2015 “Whole-body tissue stabilization and selective extractions via tissue-hydrogel hybrids for high-resolution intact circuit mapping and phenotyping” Nature protocols

We thank the reviewer for this very important remark and performed additional experiments to address these subjects. A comparison of the respective bleaching protocols (heme elution, H₂O₂, Sudan Black B, no bleaching)^{8,9} in baseline hearts was now conducted (new **Supplementary Fig. 2** and new **Supplementary Table 2**). Similar to our findings in the original manuscript, no bleaching resulted in low laser penetration depth in 488 and 561 laser lines, while bleaching with H₂O₂ gave a homogenous signal intensity in all channels imaged (new **Supplementary Fig. 2a** and **d**). We found the proposed Sudan Black protocol rather not suitable for imaging of whole hearts with LSM due to the remaining black colorization of the tissue (new **Supplementary Fig. 2b**). In the original study, Treweek et al. generated slices of brain tissue for analysis by confocal microscopy rather than by whole organ analysis. For heme elution bleaching, we applied a CUBIC1 incubation step for 2 days, as proposed in the literature⁹, after tissue fixation and before further dehydration and clearing in ECi. We obtained cleared hearts with homogenous signal intensity in 488 nm and 561 nm laser lines. However, more spots of high autofluorescence than in the BALANCE-treated hearts were detectable. These were most prominent in the left ventricular wall and intraventricular septum (new **Supplementary Fig. 2c**). Of notice, the CD31 signal intensity in the heart treated with heme elution was slightly reduced compared to the BALANCE protocol. Overall, BALANCE generated very good results as compared to other previously forwarded protocols and required significantly less time (5 hours vs. 2 days). We have added a statement for the comparison of the two other protocols:

To the revised manuscript (page 4, lines 99 – 102):

*“We tested other established protocols⁹ for tissue autofluorescence reduction and found BALANCE to be the most suitable for homogenizing tissue autofluorescence fast and throughout the whole heart using our I/R injury protocol workflow (**Supplementary Fig. 2** and **Supplementary Table 2**).”*

To the supplementary information (pages 17-18, lines 22 – 31):

“Alternative bleaching protocols

*We tested the efficiency of BALANCE against Sudan Black bleaching and heme elution, both established protocols for reducing tissue autofluorescence⁹. Sudan Black bleaching was conducted before dehydration of the sample. However, incubation with Sudan Black left the heart tissue stained black, preventing imaging deeper layers of the organ (**Supplementary Fig. 2b**). For heme elution, we incorporated CUBIC-1 reagent⁶ incubation steps before the original ECI protocol’s⁸ dehydration. This extended the time needed until the imaging process by 2 d (compared to the BALANCE protocol). We achieved autofluorescence homogenization of some parts of the heart, while we still found several spots of high background signal (**Supplementary Fig. 2c**).”*

New **Supplementary Figure 2** Tissue autofluorescence homogenization using different bleaching protocols. Evaluation of ECI clearing combined with (a) no bleaching⁸, (b) Sudan Black bleaching⁹, (c) heme elution (CUBIC 1)^{6,9} and (d) H₂O₂-based bleaching (BALANCE) with regard to signal tissue penetration in indicated fluorescent channels. Scale bar values in μm. (ECi – ethyl cinnamate, Ex. – excitation, Em. – emission)

Supplementary Table 2: Comparison of alternative bleaching approaches with the BALANCE protocol. No bleaching⁸, Sudan Black bleaching⁹ and heme elution^{6,9} were conducted as previously published.

	Autofluorescence homogenization	Sample clarity	Endogenous fluorophore bleaching	Fluorescent labeling
No bleaching⁸	no homogenization in 488 and 561	grid lines hardly visible	no	preserved
Sudan Black⁹	no homogenization in 488, 561 and 647	grid lines not visible	N.A.	preserved
Heme elution^{6,9}	homogenous in all channels; autofluorescent spots	grid lines visible; brown color	no	reduced
BALANCE	homogenous in all channels	grid lines visible; reduced color	yes	preserved

-Part of the method section are highly detailed e.g. the anaesthetics used, while the more novel and important parts of the i.e. the staining is not described in detail to be able to replicate. E.g. which vein was injected, the tail vein and what volume and which concentrations are the antibodies in?

We have added this information now to our methods section (page 12, lines 303 - 305):

“For light sheet experiments, mice received intravenous (i.v.) tail vein injections 10 min before sacrifice with 5 µg of each antibody in PBS in a total volume of 150 µl per animal of anti-mouse CD31 [...]”

Could this be done with a lectin-Alexa647 or lectin-biotin and added strep-Alexa647 instead of targeting CD31, which may be less expensive?

In general, it may be possible to detect lectin conjugated to an Alexa Fluor dye. In terms of laboratory economics, we calculated rather increased costs using this approach (always considering our available suppliers). CD31 antibody costs range below those of lectins. This is mainly due to much higher amounts (100 - 400 µg per animal) of lectins needed for our purposes (I/R injury protocol) than CD31 (5 µg per animal, average weight 25.2 ± 1.5 g) that seems to be needed to sufficiently stain the endothelium.¹⁰⁻¹² However, it might be interesting to use lectins, since different lectins have been shown to stain different parts of the hearts preferentially, at least in zebrafish.¹³

We list below examples available to us, although we acknowledge that this might differ from country to country due to suppliers, taxes etc.:

Preconjugated antibodies e.g. anti-mouse CD31 Mec13.3 AF647 antibody from Biolegend, costs amount to 165 € per 100 µg (approx. 8 € per mouse). When using purified antibody (277 € for 500 µg from BD, as mentioned in the manuscript) and conjugating this with the cited AF790 coupling kit (341 € for 5 x 100 µg), this would amount to only 6.15 € per animal. In the available literature¹⁰⁻¹², investigators used 100 - 400 µg of lectin (*Lycopersicon esculentum* (LEA) lectin or *Griffonia simplicifolia* (GSA) lectin) per animal. Sigma provides LEA lectins coupled to FITC (1 mg) for 179 €, e.g. for at least 17.9 € per animal. EYlabs sells GSA lectin conjugated to FITC for around 233 € (265 US\$) for 1 mg, i.e. at least around 23.4 € (26.5 US\$) per animal.

We have added a statement to the discussion of our manuscript (page 9, lines 234 - 237):

“It is tempting to speculate that, instead of using CD31 antibody labeling to stain the vasculature, a lectin-based approach would obtain comparable results, depending on the investigator’s choice.”¹⁰⁻¹²

- Since the Ethyl Cinnamate-based clearing procedure is new, I would recommend mentioning Ethyl Cinnamate in the abstract, so the reader can quickly identify the clearing method and differentiate it from other solvent-based methods, which would also promote the previous work of the authors.

We have implemented this point in the abstract now (page 2, line 41 - 45).

“Here we present the first three-dimensional (3D), simultaneous quantitative investigation of key I/R injury components by combining bleaching-augmented solvent-based non-toxic clearing (BALANCE) using ethyl cinnamate (ECi) with light sheet fluorescence microscopy.”

- **Abstract:**

The following sentence is quite convoluted. Consider splitting it up:

“We discover and 3D-quantify distinguishable acute and late vascular I/R damage zones containing highly localized and spatially structured neutrophil infiltrates that are modulated upon cardiac healing. “

We have changed the marked sentence accordingly. (page 2, lines 46-49 and below)

“We discover and 3D-quantify distinguishable acute and late vascular I/R damage zones. These contain highly localized and spatially structured neutrophil infiltrates that are modulated upon cardiac healing.”

- **The abbreviations CD31dim, CD31neg and CD31curly have not been explained nor defined in the text. Similar for Ly-6Gpos , TTCneg , F4/80pos**

We have now included the explanation of the abbreviations in both text and figure legends.

- **“Interestingly, at d5 the majority of neutrophils was no longer associated with the border of the residual infarct-body but localized in the CD31curly volume (Supplementary Video 6). “ How many hearts were used to verify this and similar findings? It is unclear in the text.**

We have always used 3-5 mice to verify our findings. The exact numbers are now included in the text, figure and movie legends (page 8, lines 212 - 215):

“Interestingly, at d5, the majority of neutrophils was no longer associated with the border of the residual infarct-body but localized in the CD31^{curly} volume (Fig. 4d, n=5, and Supplementary Video 6). At this time, we could also detect F4/80^{pos} macrophages throughout this area (Fig. 4e, n=5).”

- **A minor note is that the combination of green and red images in figure 2b+c and 3b are not ideal for colour-blind readers. Replacing the red with e.g. magenta would solve this.**

We apologize for the suboptimal color combination in our figures and thank the reviewer for making us aware of this. We have changed all affected figures accordingly. (**Supplementary Fig. 7, Fig. 3b**)

- Consider changing the title of the sub section in methods: “Light sheet tissue processing and clearing“ into something more appropriate, like “BALANCE protocol for tissue clearing”

We have changed the title in the method’s section according to the reviewer’s suggestion (page 13, line 345).

- Last sentence in this section: What is the concentration of ethyl cinnamate?

Ethyl cinnamate is used as a pure (99% according to the manufacturer) solution. This information has been added to the manuscript (page 14, line 358 - 362):

“After a washing step of at least 4 h in 100% ethanol, samples were warmed to RT for 5 min before transfer into 7 ml pure (99%) ethyl cinnamate (ECi, CatNo. 112372, Sigma Aldrich) in a glass vial for at least 4 h prior to imaging.”

References

- 1 Botker, H. E. *et al.* Practical guidelines for rigor and reproducibility in preclinical and clinical studies on cardioprotection. *Basic Res Cardiol* **113**, 39, doi:10.1007/s00395-018-0696-8 (2018).
- 2 Lugo-Hernandez, E. *et al.* 3D visualization and quantification of microvessels in the whole ischemic mouse brain using solvent-based clearing and light sheet microscopy. *J Cereb Blood Flow Metab* **37**, 3355-3367, doi:10.1177/0271678x17698970 (2017).
- 3 Orlich, M. & Kiefer, F. A qualitative comparison of ten tissue clearing techniques. *Histol Histopathol* **33**, 181-199, doi:10.14670/HH-11-903 (2018).
- 4 Yu, T., Qi, Y., Gong, H., Luo, Q. & Zhu, D. Optical clearing for multiscale biological tissues. *J Biophotonics* **11**, doi:10.1002/jbio.201700187 (2018).
- 5 Belle, M. *et al.* Tridimensional Visualization and Analysis of Early Human Development. *Cell* **169**, 161-173 e112, doi:10.1016/j.cell.2017.03.008 (2017).
- 6 Susaki, E. A. *et al.* Advanced CUBIC protocols for whole-brain and whole-body clearing and imaging. *Nat Protoc* **10**, 1709-1727, doi:10.1038/nprot.2015.085 (2015).
- 7 Tomer, R., Ye, L., Hsueh, B. & Deisseroth, K. Advanced CLARITY for rapid and high-resolution imaging of intact tissues. *Nat Protoc* **9**, 1682-1697, doi:10.1038/nprot.2014.123 (2014).
- 8 Klingberg, A. *et al.* Fully Automated Evaluation of Total Glomerular Number and Capillary Tuft Size in Nephritic Kidneys Using Lightsheet Microscopy. *J Am Soc Nephrol* **28**, 452-459, doi:10.1681/ASN.2016020232 (2017).
- 9 Treweek, J. B. *et al.* Whole-body tissue stabilization and selective extractions via tissue-hydrogel hybrids for high-resolution intact circuit mapping and phenotyping. *Nat Protoc* **10**, 1860-1896, doi:10.1038/nprot.2015.122 (2015).
- 10 Nguyen, C. *et al.* Near-infrared fluorescence imaging of mouse myocardial microvascular endothelium using Cy5.5-lectin conjugate. *J Biophotonics* **5**, 754-767, doi:10.1002/jbio.201100119 (2012).
- 11 Robertson, R. T. *et al.* Use of labeled tomato lectin for imaging vasculature structures. *Histochem Cell Biol* **143**, 225-234, doi:10.1007/s00418-014-1301-3 (2015).
- 12 Laitinen, L. Griffonia simplicifolia lectins bind specifically to endothelial cells and some epithelial cells in mouse tissues. *Histochem J* **19**, 225-234 (1987).
- 13 Manalo, T. *et al.* Differential Lectin Binding Patterns Identify Distinct Heart Regions in Giant Danio (*Devario aequipinnatus*) and Zebrafish (*Danio rerio*) Hearts. *Journal of Histochemistry & Cytochemistry* **64**, 687-714, doi:10.1369/0022155416667928 (2016).

REVIEWERS' COMMENTS:

Reviewer #1 (Remarks to the Author):

The authors have been highly responsive to the suggestions and comments provided in the original review.

The authors have provided new information and data. The paper has been extensively revised and has been significantly improved.

I have no further comments or questions for the authors.

Reviewer #2 (Remarks to the Author):

The authors were very responsive to the reviewer's comments, and the response were comprehensive. However, the revised manuscript would be further strengthened with the following comments:

Despite the TABLE 3, it remains unclear whether BALANCE is superior to other optical clearing methods.

Both the light-sheet imaging system and BALANCE have been previously published. The I/R-mediated changes require intense post-imaging segmentation. The I/R-induced vascular and myocardial injury seems confirmatory to the existing knowledge of I/R injury in the myocardium. Please articulate the novelties in this revised manuscript.

Reviewer #3 (Remarks to the Author):

Thank you for the revision. The issues have been addressed and I have no further concerns.

Response to the reviewers' comments regarding

Contemporaneous 3D characterization of acute and chronic myocardial I/R injury and response

We thank the editor and the reviewers for their kind consideration of our revised submission. The remaining concerns have been addressed in a point-by-point manner below:

Reviewer #1 (Remarks to the Author):

The authors have been highly responsive to the suggestions and comments provided in the original review.

The authors have provided new information and data. The paper has been extensively revised and has been significantly improved.

I have no further comments or questions for the authors.

We thank the reviewer for the appreciation of our submission and the valuable input and support to improve the key objectives of our studies.

Reviewer #2 (Remarks to the Author):

The authors were very responsive to the reviewer's comments, and the responses were comprehensive.

We thank the reviewer for his/her kind feedback and the opportunity to address the remaining remarks below

However, the revised manuscript would be further strengthened with the following comments:

Despite the TABLE 3, it remains unclear whether BALANCE is superior to other optical clearing methods.

We apologize for not clearly articulating the advantages of our protocol in our manuscript. To put more emphasis on the benefits of the BALANCE protocol, we have re-worked the discussion section of the re-revised manuscript (pages 9 - 10, lines 241 – 259 and below).

„We developed BALANCE for the fast 3D characterization of damage- and response-mechanisms in whole heart specimens. The ability to reconstruct the 3D shape of affected zones might require to re-define the current 2D-concept of myocardial infarction parameters into Volumes At Risk (VAR)/infarct-bodies to give credit to their real 3D nature.

The established protocol combines the use of non-toxic reagents, short handling and protocol times, less complexity and superior signal homogenization, required for myocardial I/R analysis and not available from other published clearing methods (Supplementary Table 1). In contrast to iDISCO clearing, ethanol was used for sample dehydration to avoid high methanol toxicity and reduce possible incompatibilities with following histological antibody stainings¹. Also, long clearing times, e.g. in CUBIC² or passive CLARITY³, were circumvented by the use of non-toxic ECi (Supplementary Table 2 and 1). BALANCE uses a minimum of hands-on time and procedure steps to circumvent initial adaption difficulties as compared to SWITCH⁴. Most importantly, it homogenizes tissue autofluorescence throughout highly autofluorescent organs such as the heart, which is not the case in the original ECi protocol⁵. With the advent of robotics and artificial intelligence (AI) to automate clearing procedures and data analyses⁵, the usability and reliability of clearing methods and staining results will be further improved.“

Both the light-sheet imaging system and BALANCE have been previously published. The I/R-mediated changes require intense post-imaging segmentation. The I/R-induced vascular and myocardial injury seems confirmatory to the existing knowledge of I/R injury in the myocardium. Please articulate the novelties in this revised manuscript.

We thank the reviewer for mentioning this and for giving us the opportunity to articulate the respective novelties in the manuscript. Our previously reported ECi-based clearing has been substantially modified to generate the favorable results in the heart presented here and this workflow has been introduced as BALANCE in this current study for the first time. As such, BALANCE is novel and has not been previously published.

The used light-sheet microscope has indeed been employed in scientific studies multiple times and by many different groups, since it is a commercial imaging system. However, the current study is not focusing on the use of a specific imaging system but rather the development of a novel approach for making I/R injury amenable to 3D imaging/analysis. The hearts prepared by this approach could be imaged on any imaging system that has the capability to handle large samples.

While we do certainly confirm existing knowledge on the impact of I/R injury we also clearly advance the field with novel insights. To the best of our knowledge no previous study has defined the 3D structure and volume of AARs or infarcts, no study has identified CD31^{curly} regions as potential proxies for previous I/R damage and also no study has provided comparably detailed analyses on the quantitative and qualitative structure of vascular systems in the normal and injured heart muscle as well as the spatial positioning of infiltrated myeloid immune cells in these areas. These are all novel and pathophysiologically relevant insights into the biology of I/R injury.

However, to further respond to the reviewer's comments we have adapted the following statement, which we had previously placed in the supplement section and which is now located in the main manuscript to articulate the BALANCE/light-sheet related features (page 10, lines 260 - 273 of the re-revised manuscript):

„Regarding time and expense of experiments, we compared the conventionally applied methods in the field of I/R injury in the murine heart with our current LSFM-based approach (Supplementary Table 3). The hands-on time of our proposed LSFM procedure is comparable to each of the classical methods⁶. However, none of the currently available approaches, including TTC staining, histology, immuno-histology and flow cytometry provides LSFM-comparable data. LSFM combines spatially resolved I/R injury parameter assessment with cellular quantification and 3D localization in the same heart. Furthermore,

LSFM-based analysis offers, inherent from its 3D nature, the possibility to reconstruct the fine ultrastructure of the inspected feature such as the blood capillary network. We therefore believe that with our proposed method, the robustness over experiments can be increased. At the same time, combined experimental times needed to carry out all other individual methods in different animals can be significantly reduced by our multiplexing approach. In the near future, computer-based algorithms will further reduce analysis time.“

Regarding the key features of our I/R injury assessment approach we have added the following statement to the discussion of our manuscript (page 11, lines 288 - 304 of the revised version of the manuscript).

„With our proposed workflow, we provide a tool for the analysis of conventional I/R injury in conjunction with the characterization of the vascular injury several days after the initial I/R event. Another problem of previously forwarded potentially cardioprotective therapies appears to be the lack of rigor in experimental studies⁶. To establish rigor in our model, we performed benchmarking. Hence we have conducted correlation analyses of our proposed analysis tool for the CD31^{neg} I/R injury against recently recommended standard approaches for experimental animal studies⁶ and show a very good correlation (Figure 4)⁶⁻¹⁰. Of note, instead of using CD31 antibody labelling to stain the vasculature, a lectin-based approach would obtain comparable results, depending on the investigator's choice¹¹⁻¹³.

Remarkably, the vasculature has been largely affected as shown by our study and the vascular injury correlates with tissue damage as assessed by classical methods. Hence, we believe that BALANCE/LSFM might also help to identify additional potential novel therapeutic options for the treatment of I/R injury in the future.“

- 1 Belle, M. *et al.* Tridimensional Visualization and Analysis of Early Human Development. *Cell* **169**, 161-173 e112, doi:10.1016/j.cell.2017.03.008 (2017).
- 2 Susaki, E. A. *et al.* Advanced CUBIC protocols for whole-brain and whole-body clearing and imaging. *Nat Protoc* **10**, 1709-1727, doi:10.1038/nprot.2015.085 (2015).
- 3 Tomer, R., Ye, L., Hsueh, B. & Deisseroth, K. Advanced CLARITY for rapid and high-resolution imaging of intact tissues. *Nat Protoc* **9**, 1682-1697, doi:10.1038/nprot.2014.123 (2014).
- 4 Yu, T., Qi, Y., Gong, H., Luo, Q. & Zhu, D. Optical clearing for multiscale biological tissues. *J Biophotonics* **11**, doi:10.1002/jbio.201700187 (2018).
- 5 Klingberg, A. *et al.* Fully Automated Evaluation of Total Glomerular Number and Capillary Tuft Size in Nephritic Kidneys Using Lightsheet Microscopy. *J Am Soc Nephrol* **28**, 452-459, doi:10.1681/ASN.2016020232 (2017).
- 6 Botker, H. E. *et al.* Practical guidelines for rigor and reproducibility in preclinical and clinical studies on cardioprotection. *Basic Res Cardiol* **113**, 39, doi:10.1007/s00395-018-0696-8 (2018).
- 7 Hendgen-Cotta, U. B. *et al.* Nitrite reductase activity of myoglobin regulates respiration and cellular viability in myocardial ischemia-reperfusion injury. *Proc Natl Acad Sci U S A* **105**, 10256-10261, doi:10.1073/pnas.0801336105 (2008).
- 8 Luedike, P. *et al.* Cardioprotection through S-nitros(yl)ation of macrophage migration inhibitory factor. *Circulation* **125**, 1880-1889, doi:10.1161/CIRCULATIONAHA.111.069104 (2012).
- 9 Rassaf, T. *et al.* Circulating nitrite contributes to cardioprotection by remote ischemic preconditioning. *Circ Res* **114**, 1601-1610, doi:10.1161/CIRCRESAHA.114.303822 (2014).
- 10 Totzeck, M., Hendgen-Cotta, U. B., French, B. A. & Rassaf, T. A practical approach to remote ischemic preconditioning and ischemic preconditioning against myocardial ischemia/reperfusion injury. *J Biol Methods* **3**, doi:10.14440/jbm.2016.149 (2016).

- 11 Nguyen, C. *et al.* Near-infrared fluorescence imaging of mouse myocardial microvascular endothelium using Cy5.5-lectin conjugate. *J Biophotonics* **5**, 754-767, doi:10.1002/jbio.201100119 (2012).
- 12 Robertson, R. T. *et al.* Use of labeled tomato lectin for imaging vasculature structures. *Histochem Cell Biol* **143**, 225-234, doi:10.1007/s00418-014-1301-3 (2015).
- 13 Laitinen, L. Griffonia simplicifolia lectins bind specifically to endothelial cells and some epithelial cells in mouse tissues. *Histochem J* **19**, 225-234 (1987).

Reviewer #3 (Remarks to the Author):

Thank you for the revision. The issues have been addressed and I have no further concerns.

We thank the reviewer for the positive feedback and suggestions, which strengthened our manuscript.